# MirrorCheck: Efficient Adversarial Defense for Vision-Language Models

## Abstract

Vision-Language Models (VLMs) are becoming increasingly vulnerable to adversarial attacks as various novel attack strategies are being proposed against these models. While existing defenses excel in unimodal contexts, they currently fall short in safeguarding VLMs against adversarial threats. To mitigate this vulnerability, we propose a novel, yet elegantly simple approach for detecting adversarial samples in VLMs. Our method leverages Text-to-Image (T2I) models to generate images based on captions produced by target VLMs. Subsequently, we calculate the similarities of the embeddings of both input and generated images in the feature space to identify adversarial samples. Empirical evaluations conducted on different datasets validate the efficacy of our approach, outperforming baseline methods adapted from image classification domains. Furthermore, we extend our methodology to classification tasks, showcasing its adaptability and model-agnostic nature. Empirical findings also show the resilience of our approach against adaptive attacks, positioning it as an excellent defense mechanism for real-world deployment against adversarial threats.

## 1 Introduction

Vision-Language Models (VLMs) have emerged as transformative tools at the intersection of computer vision (CV) and natural language processing (NLP), revolutionizing the landscape of multimodal understanding. These models hold immense importance due to their unparalleled capacity to bridge the semantic gap between visual and textual modalities (Bao et al., 2023a;b; Li et al., 2022; 2023b; Guo et al., 2023; Zhu et al., 2023; Liu et al., 2023; Li et al., 2023a), enabling machines to comprehend and generate content across modalities with human-like fluency.

However, while VLMs have demonstrated remarkable capabilities across various tasks, their robustness against adversarial attacks remains a critical concern. Recent studies (Xu et al., 2018; Li et al., 2019a; Zhang et al., 2022a; Zhou et al., 2022; Zhao et al., 2023; Yin et al., 2023; Wang et al., 2024b) have highlighted the susceptibility of VLMs to subtle variations in their input data, particularly in scenarios involving multimodal interactions. Adversaries can exploit weaknesses in VLMs by crafting imperceptible modifications to the input data that yield erroneous outputs. The interactive nature of VLMs, especially in image-grounded text generation tasks, further amplifies their vulnerability, raising concerns about their deployment in safety-critical environments (Vemprala et al., 2023; Park et al., 2023). Therefore, the need for effective defense mechanisms to safeguard against such threats is paramount.

To counter these threats in neural networks, advances have emerged in several major forms: (i) **Detectors** (Metzen et al., 2017b; Grosse et al., 2017; Feinman et al., 2017; Roth et al., 2019; Xu et al., 2017; Meng & Chen, 2017; Metzen et al., 2017a; Deng et al., 2021), designed to discern adversarial examples from natural images, (ii) **Purifiers** (Nie et al., 2022; Samangouei et al., 2018; Ho & Vasconcelos, 2022; Das et al., 2018; Hwang et al., 2019), which aim to remove adversarial features from samples, and (iii) **Ensembles** combining both detection and purification methods (Meng & Chen, 2017; Tramèr et al., 2017). Other defense mechanisms include (iv) **Adversarial Training Methods** (Goodfellow et al., 2015; Kurakin et al., 2016; Tramèr et al., 2018; Madry et al., 2018), and (v) **Certified Robustness** (Cohen et al., 2019; Salman et al., 2020; Carlini et al., 2023). However, while sophisticated detectors, purifiers, and ensemble approaches can be circumvented by knowledgeable attackers who exploit weaknesses in these defense mechanisms (adaptive attacks) Athalye et al.

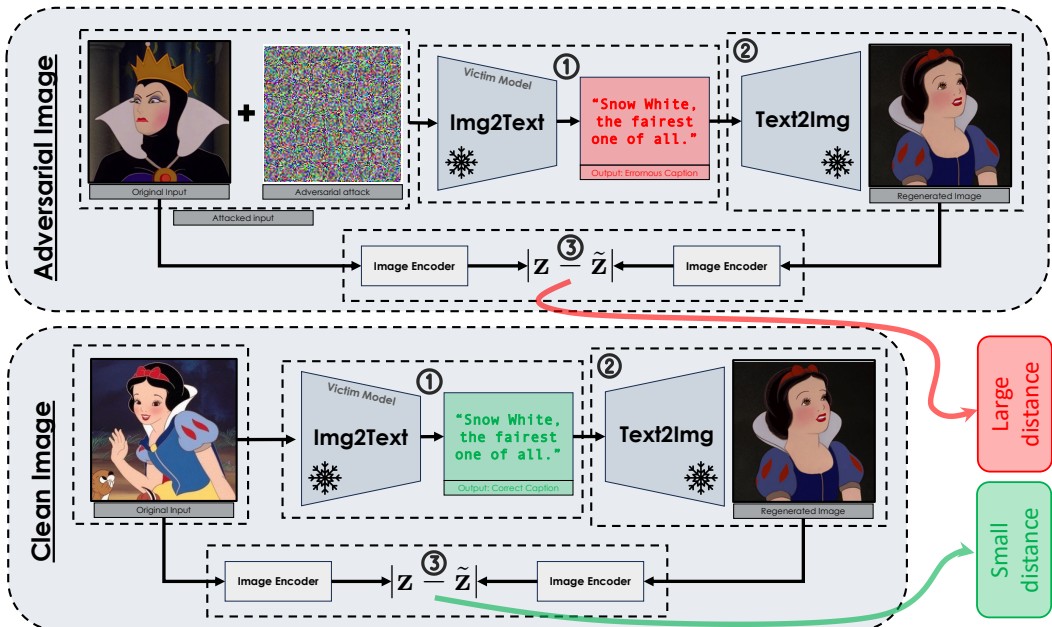

Figure 1: **MirrorCheck approach.** At inference time, to check if an input image has been adversarially attacked, our framework follows this procedure: **(1)** generates the text description for the image. **(2)** use this caption to regenerate the image with a text-to-image model. **(3)** extract and compare embeddings from both the original and regenerated images using a feature extractor. If the embeddings significantly differ, the original image likely suffered an attack. The intuition behind our method is that if the input was attacked, the image and the caption would not be semantically consistent. Therefore, using the predicted caption as a prompt for image generation would result in an image that is significantly *semantically different*.

(2018a), adversarial training and certified robustness approaches are computationally expensive, though they provide better and stronger guarantees. Moreover, these popular defense strategies have predominantly been optimized for image classification tasks, and while a few adversarial training methods (Schlarmann et al., 2024; Mao et al., 2023; Wang et al., 2024a) have been proposed for VLMs, there is a gap in efficient and robust detection defenses tailored specifically for VLMs.

To address this challenge, we introduce a novel method and the first of its kind, MirrorCheck, for detecting adversarial samples in VLMs and demonstrate the effectiveness of this method on different VL tasks. MirrorCheck leverages a Text-to-Image (T2I) model to generate an image based on the caption produced by the victim model, as illustrated in Figure 1. Subsequently, it extracts and compares the embeddings of the input image and the generated image using cosine similarity, for which a low score indicates a potential attack. This approach not only addresses the limitations of existing methods but also provides a robust solution for detecting adversarial samples in VLMs. Through empirical evaluation, we show that our approach outperforms recent methods in detecting adversarial samples and resisting adaptive attacks. Further ablations prove the robustness of MirrorCheck to the choice of T2I models and image encoders. We also adapt our approach to detecting adversarial samples in image classification tasks and demonstrate its superior performance.

In summary, our work contributes to a novel model-agnostic approach for detecting adversarial attacks on VLMs. The proposed MirrorCheck does not require training and achieves excellent results in zero-shot settings. We evaluate our method for attacks on image captioning (IC), image description (ID), and visual-question answering (VQA) tasks, and also extend it to classification tasks, and observe significant improvements compared to the state-of-the-art.

## 2 BACKGROUND

### 2.1 ADVERSARIAL ATTACKS ON VLMS

The vulnerability of VLMs arises from the potential for perturbations to impact both visual and textual modalities. Initial efforts focused on specific tasks such as visual question answering (Xu et al.,

2018; Bartolo et al., 2021; Cao et al., 2022; Kaushik et al., 2021; Kovatchev et al., 2022; Li et al., 2021c; Sheng et al., 2021; Zhang et al., 2022b) and image captioning (Aafaq et al., 2021; Li et al., 2019a; 2021a; Chen et al., 2017; Xu et al., 2019), typically in white-box settings where attackers have access to model parameters. Recently, AttackVLM (Zhao et al., 2023) has addressed black-box scenarios, where adversaries manipulate models to generate targeted responses using surrogate models like CLIP (Radford et al., 2021) and BLIP (Li et al., 2022). Similarly, VLATTACK (Yin et al., 2023) and Attack-Bard (Dong et al., 2023b) generate adversarial samples by combining image and text perturbations, targeting black-box fine-tuned models. These findings highlight significant vulnerabilities in VLM deployment. Our study evaluates the efficacy of our defense method against various VLM and classification attacks, with details on classification attacks provided in Appendix A.3.

**AttackVLM Transfer Strategy (ADV-Transfer).** Given that the victim models are VLMs, Attack-VLM Zhao et al. (2023) employs an image encoder $\tilde{\mathcal{I}}_\phi(x) \to z$, along with a publicly available T2I generative model $G_\psi(t; \eta) \to x$ (e.g., Stable Diffusion Rombach et al. (2022)), to generate a target image corresponding to target caption $t^*$ (target caption the adversary expects the victim models to return). The objective is as follows:

$$\underset{\delta:\|\delta\|_\infty \leq \varepsilon}{\arg\min} \ d(\tilde{\mathcal{I}}_\phi(x_{adv}), \tilde{\mathcal{I}}_\phi(G_\psi(t^*; \eta))), \tag{1}$$

where $x_{adv} = x + \delta$. Note that the gradient information of $G_\psi$ is not necessary when optimizing the equation above using the Project Gradient Descent (PGD) attack Madry et al. (2018).

**AttackVLM Query Strategy (ADV-Query).** The success of transfer-based attacks heavily relies on how closely the victim and surrogate models align. When a victim model can be repeatedly queried with input images to receive text outputs, adversaries can use a query-based attacking strategy to estimate gradients by maximizing the text similarity as

$$\underset{\delta:\|\delta\|_\infty \leq \varepsilon}{\arg\min} \ d(\mathcal{T}_\pi(\mathcal{F}_\theta(x_{adv;p})), \mathcal{T}_\pi(t^*)), \tag{2}$$

where $\mathcal{T}_\pi(t) \to z$ is the text encoder, $\mathcal{F}_\theta(\cdot)$ is the victim model. Since AttackVLM Zhao et al. (2023) assumes black-box access to the victim models and cannot perform backpropagation, the random gradient-free (RGF) method Nesterov & Spokoiny (2017) is employed to estimate the gradients. Transfer attack-generated adversarial examples were employed as an initialization step to enhance the efficacy of query-based attacks.

## 2.2 Adversarial Defenses

**Defense Strategies for Image Classification Tasks.** Adversarial defenses in machine learning aim to protect models from malicious inputs designed to deceive them. These defenses are crucial for maintaining model integrity and reliability, especially in security-sensitive applications. Several methods have been employed to defend against adversaries in classification tasks. For example, Defensive Distillation Papernot et al. (2016b) trains a secondary model to mimic the probability output of the original model, often with softened labels. While this approach reduces sensitivity to adversarial noise, it does not entirely eliminate attack risks. Detectors (Roth et al., 2019; Xu et al., 2017; Meng & Chen, 2017; Metzen et al., 2017a; Deng et al., 2021) identify and filter out adversarial samples, though attackers can develop strategies to circumvent these defenses. Purification methods (Nie et al., 2022; Samangouei et al., 2018; Ho & Vasconcelos, 2022; Das et al., 2018; Hwang et al., 2019) remove adversarial perturbations from input data using techniques like autoencoders or denoising filters, but may also alter legitimate inputs, affecting performance. Adversarial Training Methods (Kurakin et al., 2017; Madry et al., 2018; Tramèr et al., 2018; Shafahi et al., 2019; Wong et al., 2020; de Jorge et al., 2022; Andriushchenko & Flammarion, 2020; Zhang et al., 2019; Dong et al., 2023a) augment training datasets with adversarial examples, allowing models to learn from these perturbations, while Certified Defense Methods Cohen et al. (2019); Salman et al. (2020); Carlini et al. (2023) provide mathematical guarantees of robustness. Both approaches, however, can be computationally intensive and may struggle to generalize to novel attack strategies.

**Safeguarding VLMs.** Recent studies (Zhao et al., 2023; Yin et al., 2023) reveal a surge in novel adversarial attack strategies targeting VLMs. Despite extensive exploration of adversarial defense

Figure 2: **An example using our `MirrorCheck` framework.** For both Clean and adversarial (Adv) cases, we use the BLIP model to generate captions for the given images. Stable Diffusion then generates images based on these captions. For the clean image, different image encoders show high similarity between the input image and the generated one. Conversely, when the input image is adversarial, different image encoders show low similarity.

strategies in the literature, these strategies have primarily been developed for unimodal tasks, such as image or text classification, and are not optimized to effectively safeguard VLMs. The unique challenges presented by VLMs arise from their ability to process and integrate multimodal data—visual and textual inputs—making traditional defense methods less effective. Existing defense methods often focus on a single modality and fail to account for the complex interactions between visual and linguistic data, which can be exploited by adversaries. To the best of our knowledge, a tailored defense strategy explicitly designed for VLMs remains absent. Hence, we propose `MirrorCheck`, an approach which aims to detect such samples without necessitating alterations to the model architecture or jeopardizing its performance.

## 3 METHOD

Let $\mathcal{F}_\theta(x_{\text{in}}; p) \to t$ be the victim VLM model, where $x_{\text{in}}$ is the input image which may be clean ($x_{\text{clean}}$) or adversarial ($x_{\text{adv}}$), $p$ is the input prompt, and $t$ is the resulting output caption. In certain tasks, such as image captioning or text retrieval, the input prompt $p$ may remain empty. Let $\mathcal{I}_\phi(x) \to z$ be a pretrained image encoder and let $G_\psi(t; \eta) \to x_{\text{gen}}$ denote a pretrained text-conditioned image generation model producing image $x_{\text{gen}}$.

### 3.1 THREAT MODEL

We operate under the assumption that the attacker holds only black-box access of the victim VLM model $\mathcal{F}_\theta(x_{\text{in}}; p)$. This includes no understanding of its architecture, parameters, and training methodologies. Similarly, the detection mechanism remains oblivious or indifferent to the specific methods employed by the attacker in generating adversarial examples. **The attacker's** main objectives are: to execute targeted attacks that cause the generated caption $t$ to match a predefined target response and to adhere to an adversarial constraint defined by the $l$-norm which limits the distance between $x_{\text{clean}}$ and $x_{\text{adv}}$. In other cases, the attacker may choose to execute untargeted attacks against the victim models. **The defender's** objective is to accurately identify and flag images as either adversarial or clean.

### 3.2 MIRRORCHECK PIPELINE

Illustrated in Figures 1 and 2, our algorithm is designed to identify adversarial images within VLMs. These images are specifically crafted to deceive the underlying victim VLM model $\mathcal{F}_\theta(x_{\text{in}}; p) \to t$, by adding adversarial perturbation $\delta$ to obtain $x_{\text{adv}}$ from $x_{\text{clean}}$, while keeping the perturbation within a perturbation bound $\varepsilon$. The key observation lies in the deviation of captions generated by adversarial images from the content of the input image, which is the primary objective of the attack. To tackle this, we propose a pipeline where the generated caption undergoes scrutiny by a separate generative model $G_\psi(t; \eta) \to x_{\text{gen}}$, where $t$ denotes the caption generated by the victim model, and $x_{\text{gen}}$ represents the newly generated image. Leveraging a pretrained image encoder $\mathcal{I}_\theta(x) \to z$, we evaluate the similarity between $x_{\text{in}}$ and $x_{\text{gen}}$. In scenarios involving clean images, we anticipate a high similarity,

as the generated caption accurately reflects the image content. Conversely, in cases of adversarial images, the similarity tends to be low.

Subsequently, we employ an adversarial detector $\mathcal{D}(x) \rightarrow [0, 1]$, which categorizes the image into either the "adversarial" class (1) or the "clean" class (0), with $\tau$ serving as the decision threshold parameter, i.e.

$$\mathcal{D}(x) = \begin{cases} 1, & \text{if } \text{sim}(\mathcal{I}_\phi(x), \mathcal{I}_\phi(G_\psi(\mathcal{F}_\theta(x; p); \eta))) < \tau, \\ 0, & \text{otherwise} \end{cases}.$$

The optimal value of $\tau$ is determined using the Receiver Operating Characteristic (ROC) curve analysis. Specifically, we identify the point on the ROC curve where the difference between the true positive rate (TPR) and the false positive rate (FPR) is maximized. This approach ensures a balanced trade-off between detection sensitivity and robustness, making $\tau$ an effective decision threshold for identifying adversarial samples. However, the choice of $\tau$ may vary based on the characteristics of the specific text-to-image models or pretrained image encoders used, and we recommend calibrating $\tau$ accordingly to account for variations in model behavior.

**Intuition behind image-image similarity.** Instead of directly comparing $x_{\text{in}}$ (the input image) with the generated caption $t$, we opted to calculate the similarity between $x_{\text{in}}$ and $x_{\text{gen}}$ (the newly generated image). This decision is based on evidence in the literature indicating that these models struggle with positional relationships and variations in verb usage within sentences. This suggests that VLMs may function more like bags-of-words and, consequently, which could limit their reliability for optimizing cross-modality similarity Yuksekgonul et al. (2022).

Furthermore, we selected this embedding-based similarity metric over conventional metrics like SSIM or FID because those methods may fail to capture semantic equivalence in cases where the T2I model generates a visually different image that is still semantically similar. By utilizing vector embeddings, we aim to maintain high similarity scores in such scenarios, ensuring robustness and reliability even when T2I outputs exhibit variability in their visual representation.

Recognizing the potential issue introduced by a single image encoder used for similarity assessment (i.e., if it was used to generate the adversarial samples), we propose two complementary strategies to combat this issue. One is to **employ an ensemble of pretrained image encoders**, the similarity will be calculated as follows: $\frac{1}{N}\sum_{i=1}^{N} \text{sim}(\mathcal{I}_\phi i(x), \mathcal{I}_\phi i(G_\psi(\mathcal{F}_\theta(x; p); \eta)))$ i.e., calculates the average output across multiple image encoders $(\mathcal{I}_{\phi 1}, \mathcal{I}_{\phi 2}, ..., \mathcal{I}_{\phi N})$.

Alternatively, we suggest perturbing the weights $\phi$ of the employed image encoder $\mathcal{I}_\phi$ with noise $\gamma$ to create a **One-Time-Use Image Encoder** (OTU) so in this case $\hat{\phi} = \phi + \gamma$. This process must ensure that despite the introduced noise, the encoder's weights remain conducive to similarity evaluation. Otherwise, the weights may become corrupted, impairing the encoder's ability to assess image similarity. This strategy allows for the creation of a modified image encoder with perturbed weights, enabling one-time use for similarity evaluation while maintaining the encoder's functionality.

## 4 EXPERIMENTS

In this section, we demonstrate the effectiveness of `MirrorCheck` at detecting adversarial samples across VLM-reliant tasks. Additionally, we ablate our method in image classification tasks and explore its performances when using different T2I models and image encoders, as detailed in Appendix C. Note that all models used for our experiments are open-source to enable reproducibility. All our code and models will become publicly available.

### 4.1 IMPLEMENTATION DETAILS

We use validation images sourced from ImageNet-1K Deng et al. (2009) as the basis for clean images, which are then used to generate adversarial examples and quantitatively assess the robustness of large VLMs, following the methodology outlined in **AttackVLM** Zhao et al. (2023) and also described in Section 2. For each experiment conducted, we randomly selected 100 or 1000 images. The targeted text descriptions that were used for this purpose are also randomly chosen from MS-COCO captions Lin et al. (2014), ensuring that each clean image is paired with a corresponding descriptive prompt.

**Adversarial Setting Description.** As mentioned earlier, we followed the settings from Zhao et al. (2023). Specifically, we set the perturbation bound to $8$ and used the $l_\infty$ constraint, where the pixel values are in the range $[0, 255]$. For transfer-based attacks, we used 100-step PGD for optimization. Each step involved 100 query times for query-based attacks, and the adversarial images were updated using 8-step PGD with the estimated gradient. To further demonstrate the robustness of our method, we additionally employed the experimental setups from Attack-Bard (Dong et al., 2023b) and Attack-MMFM (Schlarmann & Hein, 2023). Detailed discussions of these two settings can be found in Appendix A.2.

**Victim models** $(\mathcal{F}_\theta(x_{\mathbf{in}}; p))$. UniDiffuser Bao et al. (2023b), BLIP Li et al. (2022), Img2Prompt Guo et al. (2023), BLIP-2 Li et al. (2023b), LLaVa Liu et al. (2023), OpenFlamingo Awadalla et al. (2023), and MiniGPT-4 Zhu et al. (2023) serve as our victim models.

**T2I models** $(G_\psi(t))$. Our primary T2I model is Stable Diffusion (SD) Rombach et al. (2022), predominantly employing the CompVis SD-v1.4 weights. In our ablation studies, we also test the UniDiffuser T2I model Bao et al. (2023b) and the ControlNet model Zhang et al. (2023) with the RunwayML SD-v1.5 weights. We run image generation for 50 time steps and generate images of $512 \times 512$ pixels in all our experiments.

**Pretrained image encoders** $(\mathcal{I}_\phi(x_{\mathbf{in}}, x_{\mathbf{gen}}))$. We use CLIP Radford et al. (2021), pretrained on OpenAI's dataset, as our primary image encoder. For the ablations, we employed OpenCLIP Ilharco et al. (2021), pretrained on the LAION-2B Schuhmann et al. (2022) dataset, and ImageNet-Pretrained Classifiers (VGG16 Simonyan & Zisserman (2014) and ResNet-50 He et al. (2016), loaded from PyTorch). Both the input images $x_{\text{in}}$ and generated images $x_{\text{gen}}$ were preprocessed using the transforms specific to these models.

Table 1: Average Similarity Scores using CLIP's image encoders to calculate the similarities between input images (Clean or Adversarial) and generated images (using Stable Diffusion). The tasks used are image captioning (IC), image description (ID), and visual question answering (VQA). **Key Takeaway**: Our method consistently observes higher similarities for clean settings than adversarial settings.

| Victim Model | Task | Setting | CLIP Image Encoders | | | | | |
| | | | RN50 | RN101 | ViT-B/16 | ViT-B/32 | ViT-L/14 | Ensemble |
| --- | --- | --- | --- | --- | --- | --- | --- | --- |
| UniDiffuser | IC | Clean | **0.718** | **0.811** | **0.755** | 0.751 | **0.713** | **0.750** |
| | | ADV-Transfer | 0.414 | 0.624 | 0.518 | **0.804** | 0.509 | 0.574 |
| | | ADV-Query | 0.408 | 0.667 | 0.537 | 0.517 | 0.533 | 0.532 |
| BLIP | IC | Clean | **0.701** | **0.805** | **0.742** | **0.728** | **0.698** | **0.735** |
| | | ADV-Transfer | 0.385 | 0.619 | 0.517 | 0.479 | 0.497 | 0.499 |
| | | ADV-Query | 0.440 | 0.675 | 0.561 | 0.531 | 0.550 | 0.551 |
| BLIP-2 | IC | Clean | **0.720** | **0.817** | **0.762** | **0.750** | **0.720** | **0.754** |
| | | ADV-Transfer | 0.430 | 0.637 | 0.543 | 0.499 | 0.526 | 0.527 |
| | | ADV-Query | 0.408 | 0.667 | 0.546 | 0.528 | 0.533 | 0.536 |
| | ID | Attack-Bard | 0.483 | 0.672 | 0.520 | 0.218 | 0.413 | 0.461 |
| Img2Prompt | VQA | Clean | **0.661** | **0.780** | **0.712** | **0.695** | **0.670** | **0.703** |
| | | ADV-Transfer | 0.380 | 0.621 | 0.508 | 0.478 | 0.501 | 0.498 |
| | | ADV-Query | 0.446 | 0.681 | 0.565 | 0.538 | 0.563 | 0.559 |
| LLaVA | VQA | Clean | **0.680** | **0.823** | **0.755** | **0.733** | **0.714** | **0.741** |
| | | Attack-MMFM | 0.539 | 0.724 | 0.626 | 0.599 | 0.596 | 0.617 |
| OpenFlamingo | VQA | Clean | **0.690** | **0.817** | **0.756** | **0.728** | **0.723** | **0.743** |
| | | Attack-MMFM | 0.535 | 0.714 | 0.618 | 0.584 | 0.609 | 0.612 |
| MiniGPT-4 | VQA | Clean | **0.624** | **0.754** | **0.681** | **0.672** | **0.647** | **0.678** |
| | | ADV-Transfer | 0.530 | 0.693 | 0.599 | 0.578 | 0.558 | 0.591 |

## 4.2 BASELINE METHODS

To our knowledge, no publicly available defense methods exist for adversarial attacks on VLMs. For comparison, we adapt existing detection approaches Meng & Chen (2017); Pu et al. (2016); Xu et al. (2017) developed for image classification. MagNet Meng & Chen (2017) uses a detector to identify adversarial inputs by evaluating their proximity to the manifold of clean images via reconstruction errors from autoencoders. Similarly, PuVAE Pu et al. (2016) employs a variational autoencoder (VAE) to project adversarial examples onto the data manifold, selecting the closest projection as the purified sample. At inference, PuVAE projects inputs to latent spaces of different class labels and uses root mean square error to identify the closest projection, thus removing adversarial perturbations.

We adapt MagNet and PuVAE for VLM attack detection by training autoencoders and VAEs on the ImageNet dataset to learn the manifold of clean images. Feature Squeeze Xu et al. (2017) creates "squeezed" versions of the input and compares model predictions on the original and squeezed inputs; significant discrepancies indicate adversarial examples. For VLMs, we adapt this by creating "squeezed" inputs and comparing the captions generated from the original and squeezed versions.

## 4.3 RESULTS

**Similarity Scores using Stable Diffusion and CLIP image encoders.** Table 1 and 6 present the average similarity scores obtained by using CLIP image encoders to extract the embeddings of input images in different settings and generate images using Stable Diffusion. The results presented in Tables 1 and 6 are based on evaluations conducted on three tasks: image captioning (IC), image description (ID), and visual question answering (VQA). Across different victim models, higher average similarity scores are consistently observed for clean images compared to adversarial ones, showing the effectiveness of our approach in adversarial sample detection. However, variations in performance among victim models reveal the differences in susceptibility to multimodal adversarial attacks. Notably, in the case of UniDiffuser, there is an instance (using ViT-B/32) where the average similarity score for transfer-based adversarial samples exceeds that of clean ones. This phenomenon occurs when the image encoder used for similarity calculation matches the one employed for generating adversarial samples for that victim model. However, our ensemble approach effectively mitigates such events by leveraging various image encoders, as shown in Figure 3, ensuring robustness against the adversarial attack strategy used.

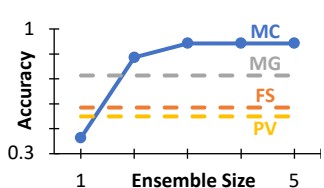

Figure 3: Effect of our ensemble approach on a victim model (Case study: UniDiffuser).

Table 2: We compare our method's detection accuracies with baseline methods; FeatureSqueeze (FS), MagNet (MN), PuVAE (PV); which were originally proposed for classification tasks. **Key Takeaway**: In the VLM domain, our method outperforms the baselines in detecting adversarial samples.

| Victim Model | Setting | CLIP Image Encoders | | | | | | Baseline Detection Methods | | |
|---|---|---|---|---|---|---|---|---|---|---|
| | | RN50 | RN101 | ViT-B/16 | ViT-B/32 | ViT-L/14 | Ens. | FS | MN | PV |
| UniDiffuser | ADV-Transfer | 0.94 | **0.96** | 0.95 | 0.39 | 0.91 | 0.92 | 0.56 | 0.74 | 0.51 |
| | ADV-Query | **0.98** | 0.95 | 0.94 | 0.93 | 0.88 | 0.97 | 0.65 | 0.85 | 0.70 |
| BLIP | ADV-Transfer | **0.90** | 0.88 | 0.84 | 0.86 | 0.80 | 0.89 | 0.52 | 0.60 | 0.50 |
| | ADV-Query | **0.89** | 0.85 | 0.75 | 0.81 | 0.73 | 0.81 | 0.57 | 0.65 | 0.80 |
| BLIP-2 | ADV-Transfer | 0.89 | **0.93** | 0.84 | 0.90 | 0.80 | 0.90 | 0.61 | 0.73 | 0.52 |
| | ADV-Query | **0.92** | 0.85 | 0.83 | 0.86 | 0.78 | 0.89 | 0.61 | 0.85 | 0.72 |
| | Attack-Bard | 0.85 | 0.84 | 0.89 | **0.98** | 0.91 | 0.89 | - | - | - |
| Img2Prompt | ADV-Transfer | **0.79** | 0.75 | 0.69 | 0.74 | 0.69 | 0.74 | 0.51 | 0.56 | 0.50 |
| | ADV-Query | 0.73 | 0.70 | 0.67 | 0.67 | 0.60 | 0.68 | - | 0.65 | **0.78** |
| LLaVA | Attack-MMFM | 0.80 | **0.82** | 0.79 | 0.78 | 0.75 | 0.79 | - | - | - |
| OpenFlamingo | Attack-MMFM | 0.76 | 0.78 | 0.79 | 0.76 | 0.75 | **0.81** | - | - | - |
| MiniGPT-4 | ADV-Transfer | 0.63 | 0.65 | 0.65 | **0.66** | **0.66** | 0.64 | 0.54 | 0.51 | 0.53 |

**Comparing `MirrorCheck` detection accuracies with baseline methods.** Using the similarity scores observed in Table 1 and 6, we compute the detection accuracies of our method under different settings. We selected the value of $\tau$ based on the validation set to maximize the difference between the true positive rate TPR (the proportion of actual adversarial images correctly identified) and the false positive rate FPR (the proportion of clean images incorrectly identified as adversarial), using various image encoders. Subsequently, we compared the performance of our method with baseline methods (FeatureSqueeze Xu et al. (2017), MagNet Meng & Chen (2017), PuVAE Hwang et al. (2019)). As illustrated in Tables 2 and 7 on both 100 and 1000 samples, our method consistently outperforms baselines in detecting both transfer-based and query-based adversarial samples. Particularly noteworthy is the performance of our CLIP-based RN50 image encoder, which outshines others across all victim models used, achieving detection accuracies ranging from 73% to 98%. Furthermore, we compared our results to a similar architecture built towards purification (DiffPure) Nie et al. (2022), and results are in the appendix (Table 5). We also provide some visualizations in Appendix D.

## 4.4 ABLATIONS

**Generalization to alternative image encoders and image generation methods.** We demonstrate the versatility of `MirrorCheck` by testing it with different Text-to-Image (T2I) models, namely UniDiffuser Bao et al. (2023a) and ControlNet Zhang et al. (2023). For this, we replace the Stable Diffusion T2I model in our framework with UniDiffuser's T2I model and ControlNet, respectively. After computing similarity scores for all configurations and victim models, we employ these scores to determine detection accuracies. Our analysis reveals an overall performance enhancement, with ControlNet delivering the most promising outcomes, as seen in Figure 4. These findings prove that `MirrorCheck` is agnostic to the T2I model and can seamlessly be combined with various generative models for image generation. We also substitute our primary image encoder, CLIP Radford et al. (2021), with alternatives from OpenCLIP Ilharco et al. (2021) and ImageNet-Pretrained Classifiers (VGG16 and ResNet-50) Simonyan & Zisserman (2014); He et al. (2016). Subsequently, we calculate similarity scores using these image models and determine detection accuracies. Refer to Appendices C.3, C.4, and C.5 for detailed results, from Tables 9 - 19.

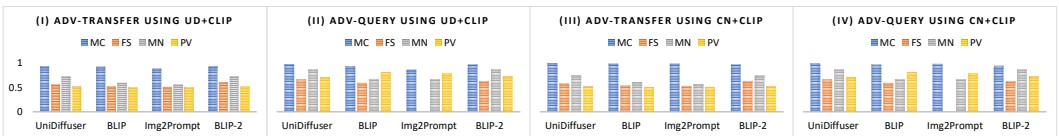

Figure 4: We carry out ablations to observe the performance of our approach, `MirrorCheck`, when we replace our baseline T2I Model (Stable Diffusion) with UniDiffuser (UD) and ControlNet (CN). We then compare our detection accuracies with baselines (Feature Squeezing (**FS** Xu et al. (2017)), MagNet (**MN**) Meng & Chen (2017), PuVAE (**PV**) Hwang et al. (2019)). Detailed results can be found in Appendices C.3, C.4, and C.5. **Key Takeaway**: Across different T2I models, `MirrorCheck` consistently surpasses all baseline methods.

**Attack Strength vs Detection Accuracy.** `MirrorCheck` is designed to be effective regardless of the attack performance of the adversarial method used. If the attack method exhibits low performance, it may fail to generate adversarial examples that meaningfully alter the model's behavior. In such cases, the robustness of our defense may not be tested to its fullest extent, but our approach will still function as intended. Specifically, our defense mechanism is built to detect discrepancies between the input and generated representations, providing reliable protection even when the adversarial perturbations are less effective. `MirrorCheck` maintains its robustness across varying levels of attack strength. As seen in Figure 5, with a very weak attack, the attack fails to generate adversarial examples that effectively alter the model's behavior. As $\epsilon$ increases, the detection accuracy improves because the adversarial perturbations become more noticeable. However, with very high values (e.g., $\epsilon = 32$), the images are almost destroyed, making them detectable even by humans.

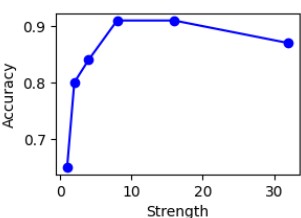

Figure 5: Attack Strength vs Detection Accuracy.

**Impact of Clean Ratio on Detection Accuracy.** We present a confusion matrix that illustrates the detection performance across different ratios of clean and adversarial examples. We observe that, as the clean ratio increases from 50% to 99.9%, the performance generally improves. This trend is particularly pronounced for the RN50 encoder, which achieves the highest ROC AUC scores, even at lower clean ratios. In contrast, encoders such as ViT-L/14 show greater sensitivity to lower clean ratios, with a notable decline in performance as the clean ratio decreases, particularly at the 99% level. This highlights that certain encoders are more robust to imbalances in clean and adversarial examples. This issue is easily solved by our ensemble approach which combines the strengths of all encoders for a well rounded performance. Interestingly, the performance stabilizes at the highest clean ratio (99.9%), where all encoders exhibit their best or near-best performances. In summary, the detection performances are stable and high regardless of the distribution of clean or

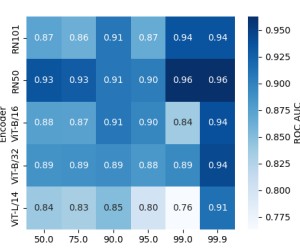

Figure 6: Attack Strength vs Detection Accuracy.

adversarial images, suggesting that our method is highly effective, even in scenarios with minimal adversarial interference.

**One-Time-Use Image Encoder Results.** By carefully applying noise to the weights, we ensure that the resulting encoder remains suitable for its intended purpose, facilitating accurate image similarity assessments even in the presence of perturbations. This method is particularly effective in restraining adaptive attacks, especially when the attacker has knowledge of the original weights. We demonstrate the effectiveness of our One-Time Use (OTU) approach using the CLIP ViT-B/32 image encoder. Detailed results, key observations, and conclusions from our experiments are presented in Appendix C.7 (from Tables 24-28).

Table 3: Robustness of `MirrorCheck` on adversarial samples generated through adaptive attacks carried out based on the attacker's knowledge of image encoders used in `MirrorCheck` pipeline. The defender employs between one and five pretrained CLIP image encoders with backbones RN50, RN101, ViT-B/16, ViT-B/32, and ViT-L/14. The attacker has knowledge of all, all but one, or all but two of these encoders, and randomly uses the remaining unknown encoders from OpenCLIP encoders.

| Attacked Image Encoder | `MirrorCheck` | | | `MirrorCheck` (OTU approach) | | |
|---|---|---|---|---|---|---|
| | ALL | ALL but ONE | ALL but TWO | ALL | ALL but ONE | ALL but TWO |
| ViT-B/32 | 0.55 | 0.90 | - | 0.50 | 0.90 | - |
| RN50 and ViT-B/32 | 0.60 | 0.70 | 0.90 | 0.70 | 0.80 | 0.90 |
| RN50, ViT-B/32, and ViT-L/14 | 0.65 | 0.65 | 0.80 | 0.75 | 0.75 | 0.80 |
| RN50, ViT-B/16, ViT-B/32, and ViT-L/14 | 0.65 | 0.65 | 0.85 | 0.75 | 0.80 | 0.85 |
| RN50, RN101, ViT-B/16, ViT-B/32, and ViT-L/14 | 0.75 | 0.75 | 0.85 | 0.85 | 0.90 | 0.80 |

**Robustness to Adaptive Attacks.** Adaptive attacks serve as a critical tool for evaluating defenses against adversarial examples, providing a dynamic and realistic assessment of a model's robustness by analyzing how attackers adapt their strategies to bypass the proposed defense. `MirrorCheck` effectively shatters the continuous gradients. Then, an attacker's objective is to generate an adversarial image ($x_{adv} = x_{in} + \delta$) by minimizing the discrepancy between its features and the target caption $t^*$, as outlined in the original attack pipeline (Adv-Transfer) Zhao et al. (2023). Moreover, the attacker aims to reduce the disparity between the features of $x_{adv}$ and the image generated from this target caption $x_{gen}$, striving for high similarity when our detection method is applied. Furthermore, the attacker will try to maintain a continuous pipeline for the entire attack, ensuring it remains differentiable. To achieve this, they can initiate the following process: starting with the victim VLM, responsible for generating the target caption $t^*$, the text embeddings of this caption obtained from the victim model text encoder $\hat{\mathcal{F}}_\theta(x; p) \to z$ are directly fed into the image generator of the attacker generative model $\hat{G}_\phi(z, \eta) \to x_{gen}$. This conditioned input generates an image closely resembling the second step of our defense mechanism. To align the text embeddings between the VLM and the generative model, the attacker must train an adapter network (MLP) capable of learning this mapping. With the entire pipeline now continuous, the attacker can perturb the input image by backpropagating through the entire process. This allows them to maximize the similarity between the adversarial image $x_{adv}$ and both the target caption $t^*$ and the generated image $x_{gen}$. This adaptive attack is illustrated in Figure 7 and Algorithm 1, where the attacker objective function is as follows:

$$\underset{\delta: \|\delta\|_\infty \leq \varepsilon}{\arg\min} \ d(\tilde{\mathcal{I}}_\phi(x_{adv}), \tilde{\mathcal{I}}_\phi(G_\psi(t^*; \eta))) + \frac{1}{N}\sum_{j=1}^{N} d(\mathcal{I}_{\phi j, \xi}(x_{adv}), \mathcal{I}_{\phi j, \xi}(\hat{G}_\psi(\mathcal{A}(\hat{\mathcal{F}}_\theta(x_{adv}; p)); \eta))). \quad (3)$$

Given that the attacker lacks knowledge about the specific image encoder $\mathcal{I}_\phi$ utilized, randomized functions and Expectation over Transformation (EOT) Athalye et al. (2018b) techniques can be employed to obtain gradients effectively. Therefore, in the adaptive part, the attacker employs multiple random image encoders, $\mathcal{I}_{\phi j, \xi}$, in an attempt to avoid detection, where $\xi$ denotes the internal randomness of the image encoder. To execute the adaptive attack technique, we vary the assumptions about the attacker's knowledge of the image encoders used in the defense pipeline. The defender employs between one and five pretrained CLIP image encoders from the following backbones: RN50, RN101, ViT-B/16, ViT-B/32, and ViT-L/14. The attacker may have knowledge of all, all but one, or all but two of these image encoders. When the defender uses more encoders than the attacker knows, the remaining unknown encoders are substituted with OpenCLIP encoders (see Appendix C.8). Additionally, we conduct experiments to show the performance of `MirrorCheck` when the defender employs the OTU approach by introducing noise into these encoders. Table 3 presents the detection accuracy for these experiments, indicating an improvement in detection accuracy when incorporating noise. The results show that using more encoders complicates the attacker's efforts to evade detection. In summary, employing multiple encoders and integrating noise both enhance robustness against adaptive attacks by increasing the difficulty of generating undetectable adversarial samples.

In our adaptive attack scenario, we consider the most challenging condition, where the attacker has full access to both the victim model and the generative model. Additionally, we explored a simpler approach for the adaptive

attack by directly using the target caption $t^*$ to construct $x_{gen}$ and search for an adversarial image $x_{adv}$. This image is optimized to simultaneously minimize the discrepancy between its features and the target caption $t^*$, as outlined in the original attack pipeline and the disparity between the features of $x_{adv}$ and $x_{gen}$. By adding the similarity measure between the adv and gen images to the original attack using the same encoder as the attack, we observed that the attack was effective, and the similarity was indeed high between the $x_{adv}$ and $x_{gen}$. However, the adversarial image was still detected by other encoders. When we employed a different encoder for optimizing the similarity between the $x_{adv}$ and $x_{gen}$, the similarity score decreased in the original attack. This led to a compromise in the main objective of the attack. We tried averaging the similarity scores across multiple encoders. However, we found that while clean images maintained high similarity scores across all encoders, the adversarial images showed variability.

**Performance of `MirrorCheck` for Attacks on Image Classification.** Although our primary focus is on Vision-Language Models (VLMs), we adapted `MirrorCheck` (MC) to match the configurations used in the baseline methods (Feature Squeezing (FS) Xu et al. (2017) and MagNet (MN) Meng & Chen (2017)), ensuring consistency in the evaluation process. Specifically, when evaluating `MirrorCheck` against Feature Squeezing, we utilized the same models (DenseNet - CIFAR10 and MobileNet - ImageNet) and adversarial attack methods (FGSM and BIM) as reported in Tables 1 and 4 of Xu et al. (2017) to maintain a fair comparison. We chose to compare `MirrorCheck`'s performance using these two attack strategies because FS reported

Table 4: Adapting `MirrorCheck` (MC) to detect adversarial samples in image classification settings.

| Dataset | Classifier | MC | FS | MN |
|---|---|---|---|---|
| CIFAR10 | DenseNet | **0.93** | 0.21 | - |
| | CNN-9 | **0.89** | - | 0.53 |
| ImageNet | MobileNet | **0.77** | 0.43 | - |

much lower performance on them compared to other utilized attacks. We hypothesize that this is due to the fact that FGSM and BIM are weaker attack strategies, making it more difficult for FS to detect adversarial features.

However, `MirrorCheck` achieved significantly better detection accuracy in both adversarial settings. We followed the same process to compare `MirrorCheck` with MN. The results for the FGSM setting ($\epsilon = 0.002$ for FS and $\epsilon = 0.01$ for MN) of both baselines, summarized in Table 4, show a significant improvement over these baselines, demonstrating the versatility of `MirrorCheck` in the detection of adversarial samples in various domains. Note that the empty cells in Table 4 correspond to methods for which results were not reported in the referenced papers. For instance, CNN-9 was only evaluated on CIFAR10 by MN, while MN did not provide results for DenseNet and MobileNet. Comprehensive reports and additional details are available in Appendix C.6, with detailed evaluations for Feature Squeezing and MagNet in Sections C.6.1 and C.6.2, respectively.

## 5 DISCUSSION

The development of Vision-Language Models (VLMs) has introduced a novel paradigm that mimics human learning from everyday sensory data. Despite their perceived robustness compared to unimodal architectures, recent literature reveals that VLMs are significantly vulnerable to new attack strategies. Recognizing this vulnerability, we introduce `MirrorCheck`, the first approach specifically tailored to detect adversarial samples in VLMs. Our extensive experiments demonstrate its efficacy across various VLM architectures and attack scenarios. Through quantitative evaluations on datasets such as ImageNet and CIFAR10, we show that `MirrorCheck` outperforms in detecting both transfer-based and query-based adversarial samples. Additionally, our method showcases robustness and adaptability, effectively functioning across different Text-to-Image (T2I) models and image encoders, underscoring its real-time practical applicability in real-world scenarios.

**Broader Impacts.** The adaptability of `MirrorCheck` extends beyond VLMs to image classification tasks, where it achieves superior detection accuracies compared to well-established methods like FeatureSqueezing. These results highlight the versatility and effectiveness of our approach in safeguarding against adversarial attacks across various machine learning tasks. By enhancing the security and robustness of VLMs and other machine learning models, `MirrorCheck` contributes to the broader field of AI safety and reliability. Its ability to detect adversarial samples in real time opens new avenues for deploying secure and resilient AI systems in diverse applications, from autonomous driving to healthcare.

**Limitations.** While `MirrorCheck` demonstrates strong performance, its effectiveness is influenced by the quality of the pretrained generative model used to generate images from the extracted captions. Any shortcomings in the generative model can directly impact the effectiveness of `MirrorCheck` in detecting adversarial samples. Future research should focus on this limitation.

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

## A  RELATED WORK

### A.1  VISUAL-LANGUAGE MODELS (VLMS)

Humans possess the remarkable ability to seamlessly integrate information from various sources concurrently. For instance, in conversations, we adeptly interpret verbal cues, body language, facial expressions, and intonation. Similarly, VLMs demonstrate proficiency in processing such multimodal signals, allowing machines to comprehend and generate image-related content that seamlessly merges visual and textual components. Contemporary VLM architectures such as CLIP Radford et al. (2021) predominantly leverage transformer-based models Vaswani et al. (2023); Dosovitskiy et al. (2021) for processing both images and text due to their effectiveness in capturing long-range dependencies. At the heart of the transformers lies the multi-head attention mechanism, which plays a pivotal role in these models' functionality.

To enable multimodal comprehension, VLMs typically comprise three key components: *(i)* an Image Model responsible for extracting meaningful visual features from visual data, *(ii)* a Text Model designed to process natural language, and *(iii)* a Fusion Mechanism to integrate representations from both modalities. Encoders in VLMs can be categorized based on their fusion mechanisms into Fusion encoders Li et al. (2020; 2021b; 2019b); Su et al. (2019), which directly combine image and text embeddings, Dual encoders Radford et al. (2021); Li et al. (2022; 2023b); Jia et al. (2021), which process modalities separately before interaction, and Hybrid methods Singh et al. (2021); Bao et al. (2022) that leverage both approaches. Furthermore, fusion schemes for cross-modal interaction can be classified into single-stream Li et al. (2020; 2019b); Su et al. (2019); Bao et al. (2022); Singh et al. (2021) and dual-stream Li et al. (2021b) architectures. The recent surge in multimodal development, driven by advances in vision-language pretraining (VLP) methods, has led to diverse vision-language applications falling into three main categories: *(i)* Image-text tasks (such as image captioning, retrieval, and visual question answering), *(ii)* Core computer vision tasks (including image classification, object detection, and image segmentation), and *(iii)* Video-text tasks (such as video captioning, video-text retrieval, and video question-answering).

### A.2  OTHER ADVERSARIAL ATTACKS USED AGAINST VLMS

**Attack-Bard (Dong et al., 2023b).** For a victim model that is a Multimodal Large Language Model (MLLM), adversarial examples that effectively perturb the image embeddings of Bard (Team et al., 2023) will consequently impact the text generation process. Let $x$ represent a natural image and $\tilde{\mathcal{I}}_{i\phi}()$ be a set of surrogate image encoders. The image embedding attack is defined as solving the following optimization problem:

$$\underset{\delta:\|\delta\|_\infty \leq \varepsilon}{\arg\max} \sum_{i=1}^{N} \|\tilde{\mathcal{I}}_{i\phi}(x_{adv}) - \tilde{\mathcal{I}}_{i\phi(x)}\|_2^2, \tag{4}$$

where $x_{adv} = x + \delta$ and the goal is to maximize the difference between the embeddings of the adversarial image $x_{adv}$ and the natural image $x$ while ensuring that the perturbation $\delta$ remains within a specified threshold $\epsilon$. To address the optimization problem in equation 4, Dong et al. (2023b) employed the SSA-CWA approach, as introduced in Chen et al. (2023).

**Attack-MMFM (Schlarmann & Hein, 2023).** An untargeted attack proposed against multimodal foundation models. To introduce minor perturbations to the visual inputs of a VLM, the authors propose a white-box untargeted attack. Specifically, given a natural image $x$, a ground truth caption $t$, along with context images $c$ and context text $z$, the objective is to design an attack that increases the negative log-likelihood of the target text $t^*$ within the constraints of the threat model:

$$\max_{\delta_x, \delta_c} - \sum_{i=1}^{m} \log p(t_i^* \mid t_{<i}^*, z, x + \delta_x, c + \delta_c) \tag{5}$$

$$\text{s.t. } \|\delta_x\|_\infty \le \epsilon_x, \|\delta_c\|_\infty \le \epsilon_c$$

In equation 5 above, $\delta_x$ is the perturbation to the input image and $\delta_c$ is the perturbation to the context images. In the setting where only the input images are attacked, optimization is performed only on $\delta_x$ and $\epsilon_c = 0$.

## A.3 ADVERSARIAL ATTACKS USED FOR CLASSIFICATION

An adversarial example, within the scope of machine learning, is a sample intentionally manipulated by an adversary to provoke an incorrect output from a target classifier. Typically, in image classification tasks, where the ground truth is based on human perception, defining adversarial examples involves perturbing a correctly classified sample (referred to as the seed example) by a limited amount to generate a misclassified sample (denoted as $x_{\text{adv}}$). Existing research on adversarial example generation predominantly centers on image classification models, reflecting the prominence and vulnerability of such models to adversarial attacks. Numerous methodologies have been introduced to craft adversarial examples, encompassing fast gradient-based techniques Goodfellow et al. (2015); Liu et al. (2016), optimization-based strategies Szegedy et al. (2013); Carlini & Wagner (2017), and other innovative approaches Nguyen et al. (2015); Papernot et al. (2016a). Notably, Carlini & Wagner (2017) introduced state-of-the-art attacks that impose constraints on $L_0$, $L_2$, and $L_\infty$ norms, highlighting the versatility and effectiveness of adversarial attacks across various norm spaces.

Adversarial examples can be categorized as targeted or untargeted depending on the adversary's objective. In targeted attacks, the adversary aims for the perturbed sample $x_{\text{adv}}$ to be classified as a specific class, while in untargeted attacks, the objective is for $x_{\text{adv}}$ to be classified as any class other than its correct class.

Formally, a targeted adversary seeks to find an $x_{\text{adv}}$ such that the target classifier assigns it to the target class $y$ while remaining within a certain distance $\epsilon$ from the original sample $x_{\text{clean}}$. Conversely, an untargeted adversary aims to find an $x_{\text{adv}}$ which is misclassified compared to the original $x_{\text{clean}}$ within the same distance threshold $\epsilon$. The adversary's strength, denoted as $\epsilon$, restricts the allowable transformations applied to the seed example. In contrast, the distance metric $\Delta(x_{\text{clean}}, x_{\text{adv}})$ and the threshold $\epsilon$ model how close an adversarial example needs to be to the original to deceive a human observer. As specified in Section 2, we will introduce some attack strategies used in classification tasks. We also leverage these attacks to test the efficacy of `MirrorCheck` in this setting;

- **Fast Gradient Sign Method** (FGSM, $L_\infty$, Untargeted): The Fast Gradient Sign Method (FGSM) is an adversarial attack technique proposed by Goodfellow et al. Goodfellow et al. (2015) that efficiently generates adversarial examples for deep neural networks (DNNs). The objective of the FGSM attack is to perturb input data in such a way that it induces misclassification by the target model while ensuring the perturbations are imperceptible to human observers. The main idea behind FGSM is to compute the gradient of the loss function with respect to the input data, and then perturb the input data in the direction that maximizes the loss. Specifically, FGSM calculates the gradient of the loss function with respect to the input data, and then scales the gradient by a small constant $\epsilon$ to determine the perturbation direction. This perturbation is added to the original input data to create the adversarial example. Mathematically, the FGSM perturbation is defined as:

$$x_{\text{adv}} = x_{\text{clean}} + \epsilon \cdot \text{sign}(\nabla_x J(w^T x_{\text{clean}}), y))$$

  where $\epsilon$ is a small constant controlling the magnitude of the perturbation, and sign denotes the sign function. The objective function of the FGSM attack is typically the cross-entropy loss between the predicted and true labels, as it aims to maximize the model's prediction error for the given input.

- **Basic Iterative Method** (BIM, $L_\infty$, Untargeted): The Basic Iterative Method (BIM) attack Feinman et al. (2017), also known as the Iterative Fast Gradient Sign Method (IFGSM), is an iterative variant of the FGSM attack designed to generate stronger adversarial examples. Like FGSM, the objective of the BIM attack is to craft adversarial perturbations that lead to misclassification by the target model while remaining imperceptible to human observers. In the BIM attack, instead of generating a single perturbation in one step, multiple small perturbations are iteratively applied to the input data. This iterative approach allows for finer control over the perturbation process, resulting in adversarial examples that are more effective and harder for the target model to defend against. The BIM attack starts with the original input data and applies small perturbations in the direction of the gradient of the loss function with respect to the input data. After each iteration, the perturbed input data is clipped to ensure it remains within a small $\epsilon$-ball around the original input. This process is repeated for a fixed number of iterations or until a stopping criterion is met. Mathematically, the perturbed input at each iteration $s$ of the BIM attack is given by:

$$x_{\text{adv}}^s = \text{clip}_\epsilon(x_{\text{adv}}^{s-1} + \alpha \cdot \text{sign}(\nabla_x J(w^T x_{\text{clean}}), y))$$

where $\text{Clip}_\epsilon$ denotes element-wise clipping to ensure the perturbation magnitude does not exceed $\epsilon$, and $\alpha$ is a small step size controlling the magnitude of each perturbation. The BIM attack aims to maximize the loss function while ensuring the perturbations remain bounded within the $\epsilon$-ball around the original input.

- **DeepFool** ($L_2$, Untargeted): The DeepFool attack Moosavi-Dezfooli et al. (2016) is an iterative and computationally efficient method for crafting adversarial examples. It operates by iteratively perturbing an input image in a direction that minimally changes the model's prediction. The objective of the DeepFool attack is to find the smallest perturbation that causes a misclassification while ensuring that the adversarial example remains close to the original input in terms of the $L_2$-norm. The DeepFool attack starts with the original input image and iteratively computes the perturbation required to push the image across the decision boundary of the model. It computes the gradient of the decision function with respect to the input and then finds the direction in which the decision boundary moves the most. By iteratively applying small perturbations in this direction, the DeepFool attack gradually moves the input image towards the decision boundary until it crosses it. Mathematically, the perturbed input at each iteration of the DeepFool attack is computed as follows:

$$x_{\text{adv}}^s = x_{\text{adv}}^{s-1} + \alpha \cdot \frac{\nabla_f(x_{\text{clean}})}{\|\nabla_f(x_{\text{clean}})\|_2}$$

where $x_{\text{adv}}^{s-1}$ is the input image at the current iteration $s$, $\alpha$ is a small step size, and $\nabla_f(x)$ is the gradient of the decision function with respect to the input image $x_{\text{clean}}$. The process continues until the model misclassifies the perturbed input or until a maximum number of iterations is reached.

- **Projected Gradient Descent** (PGD, $L_2$, Untargeted): The Projected Gradient Descent (PGD) attack Madry et al. (2018) is an advanced iterative method used for crafting adversarial examples. It builds upon the Basic Iterative Method (BIM), extending it by continuing the perturbation process until reaching a specified maximum perturbation magnitude. The objective of the PGD attack is to find the smallest perturbation that leads to misclassification while constraining the perturbed example to remain within a specified $L_p$-norm distance from the original input. The PGD attack starts with the original input image and iteratively computes the perturbation required to induce misclassification. At each iteration, it calculates the gradient of the loss function with respect to the input and applies a small step in the direction that maximizes the loss while ensuring the perturbed example remains within the specified $L_p$-norm ball around the original input. This process continues for a predetermined number of iterations or until a misclassification is achieved. Mathematically, the perturbed input at each iteration of the PGD attack is computed as follows:

$$x_{\text{adv}}^s = \text{clip}(x_{\text{adv}}^{s-1} + \alpha \cdot \text{sign}(\nabla_x J(w^T x_{\text{clean}}), y), x_{\text{adv}} - \epsilon, x_{\text{adv}} + \epsilon)$$

where $x_{\text{adv}}^{t-1}$ is the input image at the current iteration $t$, $\alpha$ is the step size, $\nabla_x J(w^T x_{\text{clean}}, y)$ is the gradient of the loss function with respect to the input image $x_{\text{clean}}$, and clip function ensures that the perturbed image remains within a specified range defined by the lower and upper bounds.

- **Carlini-Wagner** (C&W, $L_2$, Untargeted): The Carlini-Wagner (C&W) attack Carlini & Wagner (2017), introduced by Carlini and Wagner in 2017, is a powerful optimization-based method for crafting adversarial examples. Unlike many other attack methods that focus on adding imperceptible perturbations to input data, the C&W attack formulates the attack as an optimization problem aimed at finding the smallest perturbation that leads to misclassification while satisfying certain constraints. The objective of the C&W attack is to find a perturbation $\delta$ that minimizes a combination of the perturbation magnitude and a loss function, subject to various constraints. The loss function is typically designed to encourage misclassification while penalizing large perturbations. The constraints ensure that the perturbed example remains within a specified $L_p$-norm distance from the original input and maintains perceptual similarity. The objective function of the C&W attack can be formulated as follows:

$$\min \|\delta\|_l + c \cdot f(x_{\text{clean}} + \delta)$$

where $\|\delta\|_l$ represents the $L_l$-norm of the perturbation, $f(x_{\text{clean}} + \delta)$ is the loss function representing misclassification, and $c$ is a regularization parameter that balances the trade-off between the perturbation magnitude and the loss function.

# B MirrorCheck as an Autoencoder

In the auto-encoder literature, reconstruction error has been shown to be a reliable indicator of whether a sample is in or out of the training distribution Zhou (2022); Durasov et al. (2024a;b). We now cast `MirrorCheck` as a particular kind of auto-encoder to leverage these results and justify our approach. `MirrorCheck` can be conceptualized within the structure of regular Hinton & Salakhutdinov (2006); Vincent et al. (2010); Makhzani et al. (2016) and Variational Autoencoders (VAEs) (Kingma & Welling, 2014; Burda et al., 2015; Higgins et al., 2017), which typically encode input data into a continuous latent space through an encoder and reconstruct the input using a decoder. Unlike typical variational-autoencoders, `MirrorCheck` relies on a discrete, categorical latent space comprising textual descriptions generated from images. In this respect, it is in line with recent VAEs that incorporate categorical latent variables through mechanisms such as the Gumbel-Softmax distribution Maddison et al. (2016); Jang et al. (2017); Baevski et al. (2020); Sadhu et al. (2021); Gangloff et al. (2022).

The Image-to-Text phase of `MirrorCheck` acts as the encoder, mapping high-dimensional visual data into a discrete latent space represented by text. This process can be mathematically expressed as

$$q_\phi(\mathbf{z}|\mathbf{x}) = \text{Cat}(\mathbf{z}; \boldsymbol{\pi}(\mathbf{x})) , \tag{6}$$

where $\mathbf{x}$ is the input image, $\mathbf{z}$ represents the latent textual description, Cat denotes the categorical distribution, and $\boldsymbol{\pi}(\mathbf{x})$ is the distribution over the discrete latent variables conditioned on the input image, parameterized by $\phi$.

The Text-to-Image phase serves as the decoder. It reconstructs the visual data from these textual descriptions. It can be written as

$$p_\theta(\mathbf{x}|\mathbf{z}) = \text{Bernoulli}(\mathbf{x}; \boldsymbol{\sigma}(\mathbf{z})), \tag{7}$$

where $\boldsymbol{\sigma}(\mathbf{z})$ models the probability of generating an image $\mathbf{x}$ from the latent description $\mathbf{z}$, parameterized by $\theta$. When sampling caption text with a non-zero softmax temperature, these steps resemble the Gumbel-Softmax reparameterization trick, typically used in Variational Autoencoders (VAEs) to sample from the latent Maddison et al. (2016); Jang et al. (2017).

Thus, using the reconstruction error as an indication of whether an input has been compromised via an adversarial attack is as justified as using it to determine if a sample is out-of-distribution when employing a VAE. This aligns with earlier work Meng & Chen (2017); Pu et al. (2016) that showed that this metric is good at detecting adversarial attacks. It is also in the same spirit as approaches to detecting anomalies through segmentation and reconstruction Lis et al. (2019; 2024).

**Computational Efficiency.** Our experiments were carried out on a machine equipped with 80 CPUs and one NVIDIA Quadro RTX A6000 48GB GPU. The entire defense pipeline takes approximately 15 seconds per image. Within this process, obtaining a caption from the victim VLM model takes around 0.2 seconds, generating an image takes about 5 seconds, and calculating similarity requires approximately 10 seconds. However, this is the worst case scenario and there are multiple methods to improve this time i.e., reducing timesteps for generation from 50 to 10 allows the pipeline process an image in just 1.2 seconds with a little compromise in detection performance.

# C Additional Empirical Results

## C.1 Comparison with DiffPure

Table 5: Detection accuracies of DiffPure and MirrorCheck. MirrorCheck demonstrates superior performance and adaptability.

| Defense | RN50 | RN101 | ViT-B/16 | ViT-B/32 | ViT-L/14 | Ensemble |
|---------|------|-------|----------|----------|----------|----------|
| DiffPure | 0.65 | 0.61 | 0.64 | 0.62 | 0.76 | 0.65 |
| MirrorCheck (t=50) | 0.89 | 0.93 | 0.84 | 0.90 | 0.80 | 0.90 |
| MirrorCheck (t=10) | 0.87 | 0.85 | 0.83 | 0.89 | 0.78 | 0.84 |

As suggested, we conducted experiments on BLIP-2 as the victim model, with DiffPure (results shown in Table 5) and demonstrated that our method, MirrorCheck, achieves superior detection performance. Below, we outline the key differences between DiffPure and MirrorCheck, along with the results of our comparative analysis:

- **Detection vs. Purification:** DiffPure was originally designed for purification, not detection. To use DiffPure as a detection pipeline in our experiments, we passed each image through its purification pipeline and compared the embedding of the purified image to that of the original image. As shown in

Table 5, MirrorCheck consistently outperforms DiffPure in terms of detection accuracy across various model architectures.

- **Efficiency:** In our experiments, DiffPure required approximately 8 seconds per image on an RTX A6000 GPU, compared to MirrorCheck's 15 seconds per image at 50 timesteps. However, we optimized MirrorCheck by reducing the number of timesteps to 10, enabling it to process 100 images in 2 minutes (1.2 seconds per image) while maintaining a higher detection accuracy than DiffPure. This demonstrates MirrorCheck's potential for further optimization to significantly reduce processing times without substantial performance degradation.

- **Model-Agnostic Nature:** MirrorCheck is model-agnostic, meaning it is not tied to specific architectures or datasets. This flexibility makes it more difficult for attackers to create adaptive attacks against our method. Furthermore, the adaptable nature of MirrorCheck has been leveraged in other research (details withheld for blind review) to defend against jailbreaking threats. Additionally, optimizing MirrorCheck for faster performance is straightforward, as reducing the number of timesteps in the T2I model directly reduces processing time while maintaining competitive detection accuracy.

While DiffPure and MirrorCheck have different design motivations (purification vs. detection), our results show that MirrorCheck offers significant advantages in terms of detection performance and adaptability, while optimizations could boost efficiency.

## C.2 SIMILARITY SCORES AND DETECTION ACCURACIES USING 1000 IMAGES

To validate the consistency of our results on 100 images, we ran extra experiments on 1000 images. Tables 6 and 7 proves that we could get generalizable results using just 100 images.

Table 6: Similarity scores using 1000 samples for each setting. We observed similar results when using 100 images. The Min and Max similarity scores show the ranges observed on all samples used for the experiment. The average shows that `MirrorCheck` is able to maximize the difference between clean and adversarial images for all victim models.

| Victim Model | Setting | RN50 | | | RN101 | | | ViT-B/16 | | | ViT-B/32 | | | ViT-L/14 | | |
|---|---|---|---|---|---|---|---|---|---|---|---|---|---|---|---|---|
| | | Avg | Min | Max | Avg | Min | Max | Avg | Min | Max | Avg | Min | Max | Avg | Min | Max |
| UniDiffuser | Clean | **0.720** | 0.241 | 0.931 | **0.818** | 0.512 | 0.963 | **0.758** | 0.320 | 0.975 | 0.750 | 0.344 | 0.973 | **0.723** | 0.244 | 0.952 |
| | ADV-Transfer | 0.414 | 0.118 | 0.872 | 0.628 | 0.434 | 0.938 | 0.515 | 0.222 | 0.852 | **0.807** | 0.426 | 0.958 | 0.516 | 0.130 | 0.820 |
| | ADV-Query | 0.421 | 0.165 | 0.742 | 0.676 | 0.539 | 0.780 | 0.551 | 0.330 | 0.759 | 0.528 | 0.274 | 0.725 | 0.547 | 0.280 | 0.735 |
| BLIP | Clean | **0.699** | 0.162 | 0.911 | **0.804** | 0.434 | 0.953 | **0.741** | 0.247 | 0.948 | **0.723** | 0.222 | 0.945 | **0.705** | 0.126 | 0.944 |
| | ADV-Transfer | 0.395 | 0.077 | 0.823 | 0.627 | 0.455 | 0.858 | 0.522 | 0.239 | 0.847 | 0.487 | 0.173 | 0.798 | 0.512 | 0.070 | 0.828 |
| | ADV-Query | 0.443 | 0.165 | 0.694 | 0.679 | 0.522 | 0.81 | 0.563 | 0.276 | 0.740 | 0.534 | 0.212 | 0.750 | 0.561 | 0.277 | 0.757 |
| BLIP-2 | Clean | **0.712** | 0.151 | 0.936 | **0.813** | 0.422 | 0.965 | **0.757** | 0.248 | 0.961 | **0.737** | 0.213 | 0.946 | **0.725** | 0.189 | 0.948 |
| | ADV-Transfer | 0.439 | 0.045 | 0.827 | 0.644 | 0.417 | 0.884 | 0.543 | 0.218 | 0.864 | 0.498 | 0.175 | 0.844 | 0.544 | 0.140 | 0.822 |
| | ADV-Query | 0.409 | 0.124 | 0.684 | 0.668 | 0.488 | 0.791 | 0.538 | 0.316 | 0.746 | 0.519 | 0.301 | 0.721 | 0.530 | 0.249 | 0.734 |
| Img2Prompt | Clean | **0.652** | 0.212 | 0.912 | **0.775** | 0.454 | 0.946 | **0.699** | 0.297 | 0.949 | **0.684** | 0.236 | 0.93 | **0.667** | 0.151 | 0.939 |
| | ADV-Transfer | 0.389 | 0.097 | 0.798 | 0.626 | 0.426 | 0.866 | 0.517 | 0.214 | 0.822 | 0.481 | 0.161 | 0.797 | 0.508 | 0.129 | 0.794 |
| | ADV-Query | 0.448 | 0.116 | 0.698 | 0.683 | 0.501 | 0.820 | 0.564 | 0.316 | 0.731 | 0.536 | 0.240 | 0.761 | 0.563 | 0.270 | 0.801 |

Table 7: Detection accuracies using 1000 samples for each setting. TPR is the proportion of actual adversarial images that are correctly identified. FPR is the proportion of clean images incorrectly identified as adversarial. Accuracy is the proportion of correctly identified images (both clean and adversarial).

| Victim Model | Setting | RN50 | | | RN101 | | | ViT-B/16 | | | ViT-B/32 | | | ViT-L/14 | | | Ensemble | | |
|---|---|---|---|---|---|---|---|---|---|---|---|---|---|---|---|---|---|---|---|
| | | TPR | FPR | ACC | TPR | FPR | ACC | TPR | FPR | ACC | TPR | FPR | ACC | TPR | FPR | ACC | TPR | FPR | ACC |
| UniDiffuser | ADV-Transfer | 0.917 | 0.085 | **0.916** | 0.912 | 0.088 | 0.912 | 0.902 | 0.098 | 0.902 | 0.368 | 0.636 | 0.366 | 0.874 | 0.127 | 0.874 | 0.87 | 0.13 | 0.87 |
| | ADV-Query | **0.925** | 0.075 | 0.925 | 0.871 | 0.129 | 0.871 | 0.874 | 0.125 | 0.875 | 0.889 | 0.113 | 0.888 | 0.825 | 0.174 | 0.826 | 0.895 | 0.105 | 0.895 |
| BLIP | ADV-Transfer | 0.905 | 0.096 | **0.905** | 0.894 | 0.108 | 0.893 | 0.876 | 0.126 | 0.875 | 0.887 | 0.114 | 0.887 | 0.84 | 0.159 | 0.841 | 0.898 | 0.103 | 0.898 |
| | ADV-Query | 0.896 | 0.104 | **0.896** | 0.855 | 0.144 | 0.856 | 0.838 | 0.162 | 0.838 | 0.854 | 0.144 | 0.855 | 0.792 | 0.213 | 0.790 | 0.865 | 0.136 | 0.865 |
| BLIP-2 | ADV-Transfer | 0.882 | 0.119 | 0.882 | 0.883 | 0.117 | 0.883 | 0.873 | 0.128 | 0.873 | 0.898 | 0.102 | **0.898** | 0.835 | 0.166 | 0.835 | 0.891 | 0.111 | 0.890 |
| | ADV-Query | 0.921 | 0.082 | **0.920** | 0.885 | 0.117 | 0.884 | 0.886 | 0.114 | 0.886 | 0.896 | 0.104 | 0.896 | 0.856 | 0.144 | 0.856 | 0.912 | 0.090 | 0.911 |
| Img2Prompt | ADV-Transfer | 0.841 | 0.160 | **0.841** | 0.833 | 0.170 | 0.832 | 0.815 | 0.185 | 0.815 | 0.838 | 0.164 | 0.837 | 0.783 | 0.216 | 0.784 | 0.834 | 0.167 | 0.8335 |
| | ADV-Query | 0.809 | 0.195 | **0.807** | 0.759 | 0.242 | 0.7585 | 0.767 | 0.235 | 0.766 | 0.789 | 0.213 | 0.788 | 0.708 | 0.295 | 0.707 | 0.782 | 0.220 | 0.781 |

## C.3 SIMILARITY SCORES AND DETECTION ACCURACIES USING CLIP IMAGE ENCODERS

Rather than using Stable Diffusion in `MirrorCheck`, we leverage UniDiffuser T2I model Bao et al. (2023a) and ControlNet Zhang et al. (2023). **Key Takeaway**: We observe better accuracies using UniDiffuser, compared to using Stable Diffusion. We also observe better accuracies using ControlNet, compared to using Stable Diffusion, and slightly better overall accuracies compared to UniDiffuser. Tables 8 and 10 show the similarities

when using UniDiffuser-T2I Bao et al. (2023a) and ControlNet Zhang et al. (2023) for image generation and the CLIP models for evaluation, while Tables 9 and 11 show the detection accuracies.

Table 8: Similarity: UniDiffuser + CLIP.

| Victim Model | Setting | CLIP Image Encoder | | | | | |
| | | RN50 | RN101 | ViT-B/16 | ViT-B/32 | ViT-L/14 | Ensemble |
|---|---|---|---|---|---|---|---|
| UniDiffuser Bao et al. (2023b) | Clean | **0.737** | **0.826** | **0.769** | 0.764 | **0.721** | **0.763** |
| | ADV-Transfer | 0.408 | 0.617 | 0.501 | **0.765** | 0.486 | 0.555 |
| | ADV-Query | 0.396 | 0.659 | 0.526 | 0.508 | 0.520 | 0.522 |
| BLIP Li et al. (2022) | Clean | **0.713** | **0.806** | **0.742** | **0.730** | **0.685** | **0.735** |
| | ADV-Transfer | 0.375 | 0.609 | 0.500 | 0.466 | 0.480 | 0.486 |
| | ADV-Query | 0.417 | 0.656 | 0.529 | 0.503 | 0.526 | 0.526 |
| BLIP-2 Li et al. (2023b) | Clean | **0.732** | **0.823** | **0.764** | **0.759** | **0.720** | **0.760** |
| | ADV-Transfer | 0.425 | 0.627 | 0.533 | 0.491 | 0.517 | 0.519 |
| | ADV-Query | 0.390 | 0.652 | 0.511 | 0.506 | 0.510 | 0.514 |
| Img2Prompt Guo et al. (2023) | Clean | **0.663** | **0.780** | **0.703** | **0.689** | **0.660** | **0.699** |
| | ADV-Transfer | 0.369 | 0.607 | 0.494 | 0.457 | 0.474 | 0.480 |
| | ADV-Query | 0.417 | 0.656 | 0.522 | 0.502 | 0.525 | 0.525 |
| MiniGPT-4 Zhu et al. (2023) | Clean | **0.599** | **0.737** | **0.646** | **0.641** | **0.610** | **0.646** |
| | ADV-Transfer | 0.507 | 0.678 | 0.570 | 0.540 | 0.524 | 0.564 |

Table 9: Detection: UniDiffuser + CLIP.

| Victim Model | Setting | CLIP Image Encoders | | | | | |
| | | RN50 | RN101 | ViT-B/16 | ViT-B/32 | ViT-L/14 | Ensemble |
|---|---|---|---|---|---|---|---|
| UniDiffuser Bao et al. (2023b) | ADV-Transfer | **0.935** | 0.910 | 0.910 | 0.470 | 0.910 | 0.827 |
| | ADV-Query | **0.960** | 0.905 | 0.900 | 0.920 | 0.865 | 0.909 |
| BLIP Li et al. (2022) | ADV-Transfer | 0.915 | 0.910 | 0.915 | **0.920** | 0.845 | 0.901 |
| | ADV-Query | **0.920** | 0.880 | 0.900 | 0.915 | 0.820 | 0.887 |
| BLIP-2 Li et al. (2023b) | ADV-Transfer | 0.915 | 0.930 | 0.885 | **0.935** | 0.860 | 0.905 |
| | ADV-Query | **0.950** | 0.910 | 0.920 | 0.930 | 0.860 | 0.914 |
| Img2Prompt Guo et al. (2023) | ADV-Transfer | **0.885** | 0.870 | 0.830 | **0.885** | 0.810 | 0.856 |
| | ADV-Query | **0.845** | 0.810 | 0.805 | 0.830 | 0.775 | 0.813 |

Table 10: Similarity: ControlNet + CLIP.

| Victim Model | Setting | CLIP Image Encoder | | | | | |
| | | RN50 | RN101 | ViT-B/16 | ViT-B/32 | ViT-L/14 | Ensemble |
|---|---|---|---|---|---|---|---|
| UniDiffuser Bao et al. (2023b) | Clean Image | **0.747** | **0.839** | **0.768** | **0.758** | **0.731** | **0.769** |
| | ADV-Transfer | 0.410 | 0.621 | 0.514 | 0.554 | 0.514 | 0.523 |
| | ADV-Query | 0.440 | 0.663 | 0.555 | 0.522 | 0.519 | 0.540 |
| BLIP Li et al. (2022) | Clean Image | **0.747** | **0.840** | **0.770** | **0.769** | **0.728** | **0.770** |
| | ADV-Transfer | 0.398 | 0.625 | 0.526 | 0.494 | 0.511 | 0.511 |
| | ADV-Query | 0.466 | 0.689 | 0.575 | 0.527 | 0.565 | 0.564 |
| BLIP-2 Li et al. (2023b) | Clean Image | **0.751** | **0.844** | **0.774** | **0.766** | **0.735** | **0.774** |
| | ADV-Transfer | 0.388 | 0.623 | 0.526 | 0.493 | 0.512 | 0.508 |
| | ADV-Query | 0.463 | 0.684 | 0.571 | 0.522 | 0.565 | 0.561 |
| Img2Prompt Guo et al. (2023) | Clean Image | **0.661** | **0.780** | **0.712** | **0.695** | **0.670** | **0.703** |
| | ADV-Transfer | 0.400 | 0.626 | 0.532 | 0.497 | 0.514 | 0.514 |
| | ADV-Query | 0.463 | 0.685 | 0.569 | 0.534 | 0.569 | 0.564 |

Table 11: Detection: ControlNet + CLIP.

| Victim Model | Setting | CLIP Image Encoder | | | | | |
| --- | --- | --- | --- | --- | --- | --- | --- |
| | | RN50 | RN101 | ViT-B/16 | ViT-B/32 | ViT-L/14 | Ensemble |
| UniDiffuse Bao et al. (2023b) | ADV-Transfer | 0.935 | **0.980** | 0.925 | 0.895 | 0.920 | 0.931 |
| | ADV-Query | 0.945 | **0.965** | 0.880 | 0.920 | 0.880 | 0.918 |
| BLIP Li et al. (2022) | ADV-Transfer | 0.955 | **0.965** | 0.880 | 0.945 | 0.880 | 0.925 |
| | ADV-Query | **0.940** | 0.905 | 0.870 | 0.925 | 0.830 | 0.894 |
| BLIP-2 Li et al. (2023b) | ADV-Transfer | 0.935 | **0.950** | 0.905 | 0.930 | 0.900 | 0.924 |
| | ADV-Query | **0.915** | 0.910 | 0.880 | 0.890 | 0.850 | 0.889 |
| Img2Prompt Guo et al. (2023) | ADV-Transfer | **0.965** | 0.940 | 0.900 | 0.950 | 0.890 | 0.929 |
| | ADV-Query | **0.950** | 0.895 | 0.860 | 0.915 | 0.800 | 0.884 |

## C.4 SIMILARITY SCORES AND DETECTION ACCURACIES USING IMAGENET-PRETRAINED CLASSIFIERS

We calculated detection accuracies (Table 13 using similarity scores (Table 12) gotten from different T2I models (Stable Diffusion Rombach et al. (2022), UniDiffuser-T2I Bao et al. (2023a), and ControlNet Zhang et al. (2023)) and ImageNet-Pretrained Classifiers for evaluation. **Key Takeaway**: `MirrorCheck` maintains the best performances compared to baselines in Table 2.

Table 12: Similarity: (Stable Diffusion, UniDiffuser, ControlNet) + ImageNet-Pretrained Classifiers.

| Victim Model | Setting | Stable Diffusion Rombach et al. (2022) | | UniDiffuser Bao et al. (2023a) | | ControlNet Zhang et al. (2023) | |
| --- | --- | --- | --- | --- | --- | --- | --- |
| | | ResNet-50 | VGG16 | ResNet-50 | VGG16 | ResNet-50 | VGG16 |
| UniDiffuser Bao et al. (2023b) | Clean | **0.595** | **0.666** | **0.618** | **0.689** | **0.561** | **0.604** |
| | ADV-Transfer | 0.155 | 0.174 | 0.140 | 0.161 | 0.134 | 0.143 |
| | ADV-Query | 0.207 | 0.190 | 0.192 | 0.185 | 0.222 | 0.178 |
| BLIP Li et al. (2022) | Clean | **0.574** | **0.647** | **0.591** | **0.661** | **0.552** | **0.601** |
| | ADV-Transfer | 0.138 | 0.146 | 0.116 | 0.135 | 0.137 | 0.124 |
| | ADV-Query | 0.178 | 0.152 | 0.150 | 0.135 | 0.187 | 0.173 |
| BLIP-2 Li et al. (2023b) | Clean Image | **0.608** | **0.677** | **0.633** | **0.695** | **0.563** | **0.601** |
| | ADV-Transfer | 0.155 | 0.179 | 0.156 | 0.184 | 0.112 | 0.128 |
| | ADV-Query | 0.226 | 0.156 | 0.193 | 0.148 | 0.197 | 0.148 |
| Img2Prompt Guo et al. (2023) | Clean | **0.538** | **0.597** | **0.535** | **0.605** | **0.528** | **0.575** |
| | ADV-Transfer | 0.117 | 0.124 | 0.124 | 0.128 | 0.137 | 0.150 |
| | ADV-Query | 0.188 | 0.157 | 0.162 | 0.146 | 0.179 | 0.138 |

Table 13: Detection: (Stable Diffusion, UniDiffuser, ControlNet) + ImageNet-Pretrained Classifiers.

| Victim Model | Setting | Stable Diffusion Rombach et al. (2022) | | UniDiffuser Bao et al. (2023a) | | ControlNet Zhang et al. (2023) | |
| --- | --- | --- | --- | --- | --- | --- | --- |
| | | ResNet-50 | VGG16 | ResNet-50 | VGG16 | ResNet-50 | VGG16 |
| UniDiffuser Bao et al. (2023b) | ADV-Transfer | 0.830 | 0.835 | **0.885** | 0.855 | 0.855 | 0.850 |
| | ADV-Query | 0.870 | 0.910 | 0.875 | **0.920** | 0.835 | 0.855 |
| BLIP Li et al. (2022) | ADV-Transfer | **0.870** | **0.870** | 0.865 | 0.850 | 0.840 | 0.825 |
| | ADV-Query | 0.860 | 0.895 | 0.865 | **0.915** | 0.845 | 0.860 |
| BLIP-2 Li et al. (2023b) | ADV-Transfer | 0.865 | 0.865 | **0.895** | 0.880 | 0.870 | 0.845 |
| | ADV-Query | 0.855 | 0.915 | 0.910 | **0.940** | 0.830 | 0.855 |
| Img2Prompt Guo et al. (2023) | ADV-Transfer | 0.825 | 0.835 | 0.865 | **0.875** | 0.850 | 0.815 |
| | ADV-Query | 0.820 | **0.870** | 0.840 | 0.850 | 0.825 | 0.865 |

## C.5 SIMILARITY SCORES USING OPENCLIP IMAGE ENCODERS

We calculate detection accuracies for Stable Diffusion (Table 15), UniDiffuser (Table 17), and ControlNet (Table 19) using their respective similarity scores (Tables 14, 16, 18) and the OpenCLIP Ilharco et al. (2021) Image Encoders. **Key Takeaway**: We observe better overall detection accuracies on query-based adversarial samples, compared to when using ControlNet+CLIP (Table 11). Generally, `MirrorCheck` maintains its SOTA detection performance, proving that our approach is agnostic to the choice of T2I models and Image Encoders.

Table 14: Similarity: Stable Diffusion + OpenCLIP.

| Victim Model | Setting | OpenCLIP Image Encoders | | | | | |
| | | RN50 | RN101 | ViT-B/16 | ViT-B/32 | ViT-L/14 | Ensemble |
| --- | --- | --- | --- | --- | --- | --- | --- |
| UniDiffuser Bao et al. (2023b) | Clean Image | **0.525** | **0.537** | **0.618** | **0.641** | **0.579** | **0.580** |
| | ADV-Transfer | 0.218 | 0.232 | 0.296 | 0.377 | 0.253 | 0.275 |
| | ADV-Query | 0.193 | 0.177 | 0.226 | 0.296 | 0.111 | 0.200 |
| BLIP Li et al. (2022) | Clean Image | **0.505** | **0.518** | **0.598** | **0.620** | **0.551** | **0.558** |
| | ADV-Transfer | 0.209 | 0.216 | 0.272 | 0.330 | 0.235 | 0.252 |
| | ADV-Query | 0.215 | 0.196 | 0.237 | 0.311 | 0.142 | 0.220 |
| BLIP-2 Li et al. (2023b) | Clean Image | **0.512** | **0.534** | **0.628** | **0.649** | **0.591** | **0.583** |
| | ADV-Transfer | 0.221 | 0.231 | 0.294 | 0.350 | 0.265 | 0.272 |
| | ADV-Query | 0.199 | 0.175 | 0.227 | 0.309 | 0.122 | 0.207 |
| Img2Prompt Guo et al. (2023) | Clean Image | **0.450** | **0.468** | **0.543** | **0.578** | **0.494** | **0.507** |
| | ADV-Transfer | 0.201 | 0.209 | 0.266 | 0.328 | 0.226 | 0.246 |
| | ADV-Query | 0.217 | 0.199 | 0.227 | 0.306 | 0.130 | 0.216 |

Table 15: Detection: Stable Diffusion + OpenCLIP.

| Victim Model | Setting | OpenCLIP Image Encoder | | | | | |
| | | RN50 | RN101 | ViT-B/16 | ViT-B/32 | ViT-L/14 | Ensemble |
| --- | --- | --- | --- | --- | --- | --- | --- |
| UniDiffuser Bao et al. (2023b) | ADV-Transfer | **0.925** | 0.920 | **0.925** | 0.910 | 0.900 | 0.916 |
| | ADV-Query | 0.940 | 0.950 | 0.980 | 0.970 | **0.990** | 0.966 |
| BLIP Li et al. (2022) | ADV-Transfer | 0.905 | 0.925 | **0.930** | 0.895 | 0.890 | 0.909 |
| | ADV-Query | 0.905 | 0.940 | 0.955 | 0.920 | **0.975** | 0.939 |
| BLIP-2 Li et al. (2023b) | ADV-Transfer | 0.915 | 0.915 | **0.920** | 0.915 | **0.920** | 0.917 |
| | ADV-Query | 0.935 | 0.960 | **0.970** | 0.960 | **0.970** | 0.959 |
| Img2Prompt Guo et al. (2023) | ADV-Transfer | 0.830 | 0.820 | 0.885 | **0.890** | 0.810 | 0.847 |
| | ADV-Query | 0.815 | 0.835 | 0.900 | 0.880 | **0.930** | 0.872 |

Table 16: Similarity: UniDiffuser + OpenCLIP.

| Victim Model | Setting | OpenCLIP Image Encoder | | | | | |
| | | RN50 | RN101 | ViT-B/16 | ViT-B/32 | ViT-L/14 | Ensemble |
| --- | --- | --- | --- | --- | --- | --- | --- |
| UniDiffuser Bao et al. (2023b) | Clean | **0.531** | **0.547** | **0.636** | **0.659** | **0.578** | **0.590** |
| | ADV-Transfer | 0.202 | 0.209 | 0.290 | 0.371 | 0.250 | 0.264 |
| | ADV-Query | 0.190 | 0.183 | 0.228 | 0.297 | 0.104 | 0.200 |
| BLIP Li et al. (2022) | Clean | **0.512** | **0.522** | **0.596** | **0.627** | **0.539** | **0.559** |
| | ADV-Transfer | 0.189 | 0.203 | 0.271 | 0.326 | 0.232 | 0.244 |
| | ADV-Query | 0.194 | 0.183 | 0.229 | 0.293 | 0.130 | 0.206 |
| BLIP 2 Li et al. (2023b) | Clean | **0.529** | **0.539** | **0.625** | **0.652** | **0.577** | **0.584** |
| | ADV-Transfer | 0.196 | 0.208 | 0.299 | 0.353 | 0.264 | 0.264 |
| | ADV-Query | 0.178 | 0.172 | 0.223 | 0.300 | 0.112 | 0.197 |
| Img2Prompt Guo et al. (2023) | Clean | **0.447** | **0.464** | **0.532** | **0.563** | **0.469** | **0.495** |
| | ADV-Transfer | 0.185 | 0.194 | 0.252 | 0.316 | 0.207 | 0.231 |
| | ADV-Query | 0.198 | 0.189 | 0.226 | 0.293 | 0.124 | 0.206 |

Table 17: Detection: UniDiffuser + OpenCLIP.

| Victim Model | Setting | OpenCLIP Image Encoder | | | | | |
| | | RN50 | RN101 | ViT-B/16 | ViT-B/32 | ViT-L/14 | Ensemble |
| --- | --- | --- | --- | --- | --- | --- | --- |
| UniDiffuser Bao et al. (2023b) | ADV-Transfer | 0.925 | **0.940** | 0.920 | 0.900 | 0.910 | 0.919 |
| | ADV-Query | 0.940 | 0.940 | 0.975 | **0.990** | 0.980 | 0.965 |
| BLIP Li et al. (2022) | ADV-Transfer | 0.935 | **0.960** | 0.930 | 0.945 | 0.900 | 0.934 |
| | ADV-Query | 0.960 | 0.965 | 0.965 | 0.950 | **0.970** | 0.962 |
| BLIP-2 Li et al. (2023b) | ADV-Transfer | 0.930 | 0.930 | **0.945** | 0.930 | 0.910 | 0.929 |
| | ADV-Query | 0.950 | 0.960 | 0.965 | 0.965 | **0.995** | 0.967 |
| Img2Prompt Guo et al. (2023) | ADV-Transfer | 0.870 | 0.855 | **0.885** | 0.840 | 0.875 | 0.865 |
| | ADV-Query | 0.865 | 0.875 | 0.890 | 0.875 | **0.935** | 0.888 |

Table 18: Similarity: ControlNet + OpenCLIP.

| Victim Model | Setting | OpenCLIP Image Encoder | | | | | |
| | | RN50 | RN101 | ViT-B/16 | ViT-B/32 | ViT-L/14 | Ensemble |
|---|---|---|---|---|---|---|---|
| UniDiffuser Bao et al. (2023b) | Clean | **0.531** | **0.542** | **0.623** | **0.647** | **0.583** | **0.585** |
| | ADV-Transfer | 0.221 | 0.237 | 0.303 | 0.382 | 0.251 | 0.279 |
| | ADV-Query | 0.203 | 0.183 | 0.235 | 0.304 | 0.225 | 0.230 |
| BLIP Li et al. (2022) | Clean | **0.512** | **0.519** | **0.601** | **0.615** | **0.555** | **0.560** |
| | ADV-Transfer | 0.215 | 0.221 | 0.274 | 0.327 | 0.235 | 0.254 |
| | ADV-Query | 0.213 | 0.121 | 0.231 | 0.318 | 0.203 | 0.233 |
| BLIP-2 Li et al. (2023b) | Clean | **0.531** | **0.544** | **0.646** | **0.660** | **0.606** | **0.597** |
| | ADV-Transfer | 0.231 | 0.237 | 0.303 | 0.358 | 0.285 | 0.283 |
| | ADV-Query | 0.208 | 0.184 | 0.233 | 0.313 | 0.194 | 0.226 |
| Img2Prompt Guo et al. (2023) | Clean | **0.467** | **0.475** | **0.573** | **0.604** | **0.519** | **0.528** |
| | ADV-Transfer | 0.203 | 0.210 | 0.278 | 0.321 | 0.243 | 0.251 |
| | ADV-Query | 0.212 | 0.196 | 0.228 | 0.301 | 0.193 | 0.226 |

Table 19: Detection: ControlNet + OpenCLIP.

| Victim Model | Setting | OpenClip Image Encoder | | | | | |
| | | RN50 | RN101 | ViT-B/16 | ViT-B/32 | ViT-L/14 | Ensemble |
|---|---|---|---|---|---|---|---|
| UniDiffuser Bao et al. (2023b) | ADV-Transfer | 0.900 | 0.825 | 0.930 | 0.845 | **0.935** | 0.893 |
| | ADV-Query | 0.930 | 0.940 | 0.955 | 0.900 | **0.960** | 0.937 |
| BLIP Li et al. (2022) | ADV-Transfer | 0.920 | 0.960 | 0.955 | 0.945 | **0.985** | 0.953 |
| | ADV-Query | 0.925 | 0.965 | 0.970 | 0.970 | **0.980** | 0.962 |
| BLIP-2 Li et al. (2023b) | ADV-Transfer | 0.975 | **0.990** | **0.990** | 0.985 | 0.965 | 0.981 |
| | ADV-Query | **0.995** | **0.995** | 0.990 | 0.990 | 0.980 | 0.990 |
| Img2Prompt Guo et al. (2023) | ADV-Transfer | 0.950 | 0.910 | 0.935 | **0.995** | 0.945 | 0.947 |
| | ADV-Query | 0.955 | 0.925 | 0.935 | **0.995** | **0.995** | 0.961 |

## C.6 Adapting MirrorCheck for Classification Tasks

### C.6.1 MirrorCheck vs FeatureSqueezing

We implement MirrorCheck on DenseNet Iandola et al. (2014) trained on CIFAR10 and MobileNet Sandler et al. (2018) trained on ImageNet datasets, following the configurations and hyperparameters outlined in FeatureSqueeze Xu et al. (2017). The classifiers are subjected to adversarial attacks using FGSM Huang et al. (2017) and BIM Kurakin et al. (2018) strategies. To adapt MirrorCheck for comparison with FeatureSqueeze, we input $x_{in}$ into the classifier $f_\theta(\cdot)$, extract the predicted class name using argmax, and generate an image using either Stable Diffusion or ControlNet. Subsequently, we compute similarity scores and detection results. Table 20 shows classification similarities using Stable Diffusion and ControlNet for image generation and the CLIP Image Encoders for evaluation. Comparing our results in Table 21 with the best reported outcomes from various FeatureSqueezing configurations—Bit Depth, Median Smoothing, Non-Local Mean, and Best Joint Detection—we observe significant improvements. On CIFAR10 with DenseNet, our best setting achieves a detection accuracy of 91.5% against FGSM adversarial samples, compared to FeatureSqueeze's 20.8%. Similarly, for BIM samples, our approach achieves an accuracy of 87% compared to FeatureSqueeze's 55%. For ImageNet with MobileNet, our approach also outperforms FeatureSqueeze. Against FGSM samples, our best setting achieves a detection accuracy of 76.5% compared to FeatureSqueeze's 43.4%, and for BIM samples, it achieves 79.5% compared to FeatureSqueeze's 64.4%.

Table 20: Classification Similarity: (Stable Diffusion and ControlNet) + CLIP Image Encoders.

| Classifier | Setting | CLIP Image Encoder | | | | | |
|---|---|---|---|---|---|---|---|
| | | RN50 | RN101 | ViT-B/16 | ViT-B/32 | ViT-L/14 | Ensemble |
| DenseNet-CIFAR10 | Clean-CN | 0.607 | **0.761** | **0.729** | **0.697** | **0.690** | **0.697** |
| | ADV-FGSM-CN | 0.571 | 0.729 | 0.650 | 0.625 | 0.594 | 0.634 |
| | ADV-BIM-CN | **0.614** | 0.750 | 0.652 | 0.649 | 0.606 | 0.654 |
| | Clean Image-SD | **0.543** | 0.740 | 0.705 | 0.671 | 0.674 | 0.667 |
| | ADV-FGSM-SD | 0.444 | 0.666 | 0.572 | 0.537 | 0.548 | 0.553 |
| | ADV-BIM-SD | 0.507 | 0.713 | 0.593 | 0.554 | 0.532 | 0.579 |
| MobileNet-ImageNet | Clean-CN | **0.659** | **0.786** | **0.731** | **0.715** | **0.711** | **0.720** |
| | ADV-FGSM-CN | 0.578 | 0.744 | 0.632 | 0.634 | 0.599 | 0.617 |
| | ADV-BIM-CN | 0.540 | 0.718 | 0.595 | 0.601 | 0.558 | 0.602 |
| | Clean Image-SD | 0.668 | 0.790 | 0.729 | 0.704 | 0.705 | 0.719 |
| | ADV-FGSM-SD | 0.520 | 0.712 | 0.606 | 0.612 | 0.585 | 0.607 |
| | ADV-BIM-SD | 0.503 | 0.693 | 0.581 | 0.565 | 0.538 | 0.576 |

Table 21: Detection Accuracy: MirrorCheck vs FeatureSqueezing.

| Classifier | Dataset | Defense Method | Configuration | Attack Setting | |
|---|---|---|---|---|---|
| | | | | FGSM | BIM |
| DenseNet | CIFAR10 | FeatureSqueezing | Bit Depth | 0.125 | 0.250 |
| | | | Median Smoothing | 0.188 | 0.550 |
| | | | Non-Local Mean | 0.167 | 0.525 |
| | | | Joint Detection | 0.208 | 0.550 |
| | | MirrorCheck (Using SD) | RN50 | 0.795 | 0.620 |
| | | | RN101 | 0.825 | 0.595 |
| | | | ViT-B/16 | 0.860 | 0.845 |
| | | | ViT-B/32 | 0.915 | 0.850 |
| | | | ViT-L/14 | 0.850 | **0.870** |
| | | | Ensemble | **0.925** | 0.810 |
| | | MirrorCheck (Using CN) | RN50 | 0.630 | 0.450 |
| | | | RN101 | 0.650 | 0.550 |
| | | | ViT-B/16 | 0.750 | 0.725 |
| | | | ViT-B/32 | 0.740 | 0.660 |
| | | | ViT-L/14 | 0.800 | 0.760 |
| | | | Ensemble | 0.760 | 0.685 |
| MobileNet | ImageNet | FeatureSqueezing | Bit Depth | 0.151 | 0.556 |
| | | | Median Smoothing | 0.358 | 0.444 |
| | | | Non-Local Mean | 0.226 | 0.467 |
| | | | Joint Detection | 0.434 | 0.644 |
| | | MirrorCheck (Using SD) | RN50 | 0.745 | 0.755 |
| | | | RN101 | 0.680 | 0.720 |
| | | | ViT-B/16 | 0.715 | 0.775 |
| | | | ViT-B/32 | 0.710 | 0.755 |
| | | | ViT-L/14 | **0.765** | 0.785 |
| | | | Ensemble | 0.725 | **0.800** |
| | | MirrorCheck (Using CN) | RN50 | 0.685 | 0.700 |
| | | | RN101 | 0.600 | 0.690 |
| | | | ViT-B/16 | 0.725 | 0.780 |
| | | | ViT-B/32 | 0.650 | 0.725 |
| | | | ViT-L/14 | 0.750 | 0.795 |
| | | | Ensemble | 0.700 | 0.735 |

### C.6.2 MIRRORCHECK VS MAGNET

We also implement and compare `MirrorCheck` with MagNet Meng & Chen (2017). Table 22 show the classification similarities using Stable Diffusion Rombach et al. (2022) for image generation and the CLIP Image Encoders for evaluation. Subsequently, we compare `MirrorCheck` with MagNet Meng & Chen (2017) using the same settings as reported in Meng & Chen (2017). Our approach demonstrate a superior performance over MagNet. **Key Takeaway**: From experiments performed on CIFAR-10, using the classifier specified in Meng & Chen (2017), `MirrorCheck` outperforms MagNet in detecting adversarial samples in classification settings, proving the efficacy of our approach in multiple scenarios. Table 23 shows the comparison with MagNet.

Table 22: Classification Similarity: Stable Diffusion + CLIP Image Encoders.

| Setting | Eps ($\epsilon$) | CLIP Image Encoder | | | | | |
| | | RN50 | RN101 | ViT-B/16 | ViT-B/32 | ViT-L/14 | Ensemble |
|---|---|---|---|---|---|---|---|
| Clean | | **0.554** | **0.734** | **0.695** | **0.664** | **0.641** | **0.658** |
| FGSM | $\epsilon = 0.01$ | 0.456 | 0.685 | 0.574 | 0.542 | 0.532 | 0.558 |
| FGSM | $\epsilon = 0.1$ | 0.408 | 0.633 | 0.484 | 0.475 | 0.519 | 0.504 |
| L2-PGD | $\epsilon = 0.01$ | 0.488 | 0.691 | 0.613 | 0.580 | 0.563 | 0.587 |
| L2-PGD | $\epsilon = 0.5$ | 0.494 | 0.687 | 0.601 | 0.573 | 0.551 | 0.581 |
| DeepFool | $\epsilon = 0.1$ | 0.482 | 0.689 | 0.606 | 0.574 | 0.560 | 0.582 |
| C&W | $\epsilon = 0.1$ | 0.506 | 0.699 | 0.620 | 0.587 | 0.569 | 0.596 |

Table 23: Detection Accuracy: `MirrorCheck` vs MagNet (MN) Meng & Chen (2017).

| Setting | Eps ($\epsilon$) | CLIP Image Encoder | | | | | | MN Meng & Chen (2017) |
| | | RN50 | RN101 | ViT-B/16 | ViT-B/32 | ViT-L/14 | Ensemble | |
|---|---|---|---|---|---|---|---|---|
| FGSM | $\epsilon = 0.01$ | 0.660 | 0.655 | 0.770 | **0.790** | 0.750 | 0.750 | 0.525 |
| FGSM | $\epsilon = 0.1$ | 0.750 | 0.785 | **0.890** | 0.880 | 0.755 | 0.845 | 0.885 |
| L2-PGD | $\epsilon = 0.01$ | 0.635 | 0.645 | 0.730 | **0.750** | 0.715 | 0.735 | 0.490 |
| L2-PGD | $\epsilon = 0.5$ | 0.600 | 0.655 | 0.720 | **0.730** | 0.710 | 0.710 | 0.485 |
| DeepFool | $\epsilon = 0.1$ | 0.625 | 0.665 | 0.710 | **0.735** | 0.685 | 0.730 | 0.525 |
| C&W | $\epsilon = 0.1$ | 0.590 | 0.615 | **0.725** | 0.705 | 0.685 | 0.705 | 0.530 |

### C.7 MIRRORCHECK: ONE-TIME-USE (OTU) IMAGE ENCODER APPROACH

In this section, we show results from different applications of our OTU approach on the CLIP ViT-B/32 Image Encoder. Tables 24, 25, 26, 27, and 28 show the detailed descriptions of each of our experiments, along with our key observations and conclusions.

Table 24: We started by adding different pertubation values $\eta$ to the CLIP ViT-B/32 Image Encoder weights. **Key Takeaway**: Very small $\eta$ (i.e., $\eta \leq 10^{-4}$) doesn't change the model, and large $\eta$ (i.e., $\eta \geq 10^{-2}$) destroys the model's usage. This sets our optimal $\eta$ at $10^{-4} \leq \eta < 10^{-2}$.

| Victim Model | Setting | One-Time-Use (OTU) ViT-B/32 Image Encoder | | | | |
| | | $\eta = 5 \cdot 10^{-6}$ | $\eta = 3 \cdot 10^{-4}$ | $\eta = 10^{-3}$ | $\eta = 10^{-2}$ | $\eta = 10^{-1}$ |
|---|---|---|---|---|---|---|
| UniDiffuser | Clean | 0.751 | 0.752 | 0.755 | 0.867 | 0.721 |
| | ADV-Transfer | 0.804 | 0.803 | 0.778 | 0.872 | 0.717 |
| BLIP | Clean | **0.728** | **0.731** | **0.741** | 0.871 | 0.715 |
| | ADV-Transfer | 0.478 | 0.486 | 0.506 | 0.866 | 0.706 |
| BLIP-2 | Clean | **0.750** | **0.751** | **0.758** | 0.870 | 0.722 |
| | ADV-Transfer | 0.499 | 0.505 | 0.524 | 0.865 | 0.706 |
| Img2Prompt | Clean | **0.695** | **0.696** | **0.708** | 0.862 | 0.716 |
| | ADV-Transfer | 0.478 | 0.484 | 0.506 | 0.859 | 0.706 |

Table 25: To prove our conclusion in Table 24, we investigated with more perturbation values. **Key Takeaway**: Larger $\eta$ values destroy the usefulness of the encoder. Therefore, $\eta$ should be low enough.

| Victim Model | Setting | One-Time-Use (OTU) ViT-B/32 Image Encoder | | | |
| | | $\eta = 10^{-3}$ | $\eta = 5 \cdot 10^{-3}$ | $\eta = 10^{-1}$ | $\eta = 5 \cdot 10^{-1}$ |
|---|---|---|---|---|---|
| UniDiffuser | Clean | 0.755 | 0.799 | 0.721 | 0.859 |
| | ADV-Transfer | 0.778 | 0.789 | 0.717 | 0.855 |
| BLIP | Clean | **0.741** | **0.800** | 0.715 | 0.860 |
| | ADV-Transfer | 0.506 | 0.506 | 0.706 | 0.857 |

Table 26: We then investigate which layer carries the most importance for perturbing, to get to our goal of having an OTU encoder. We started by adding different pertubation values $\eta$ to the weights of the first layer (conv1) of CLIP ViT-B/32 Image Encoder. **Key Takeaway**: We observe similar trends in this case, as compared to Table 24. Our optimal $\eta$ sits at $10^{-4} \leq \eta < 10^{-2}$.

| Victim Model | Setting | One-Time-Use (OTU) ViT-B/32 Image Encoder | | | | |
| | | $\eta = 5 \cdot 10^{-6}$ | $\eta = 3 \cdot 10^{-4}$ | $\eta = 10^{-3}$ | $\eta = 10^{-2}$ | $\eta = 10^{-1}$ |
|---|---|---|---|---|---|---|
| UniDiffuser Bao et al. (2023b) | Clean | 0.751 | 0.752 | 0.749 | 0.882 | 0.881 |
| | ADV-Transfer | 0.804 | 0.802 | 0.785 | 0.865 | 0.894 |
| Blip Li et al. (2022) | Clean | **0.728** | **0.728** | **0.725** | 0.881 | 0.887 |
| | ADV-Transfer | 0.478 | 0.482 | 0.490 | 0.843 | 0.877 |
| Blip-2 Li et al. (2023b) | Clean | **0.750** | **0.749** | **0.744** | 0.881 | 0.888 |
| | ADV-Transfer | 0.499 | 0.502 | 0.511 | 0.852 | 0.883 |
| Img2Prompt Guo et al. (2023) | Clean | **0.695** | **0.694** | **0.692** | 0.875 | 0.881 |
| | ADV-Transfer | 0.478 | 0.479 | 0.489 | 0.845 | 0.876 |

Table 27: We also investigate the model's performance when perturbing the pre-weight layer ($visual.ln\_pre.weight$) of the used encoder. **Key Takeaway**: We observe a slightly different trend in this case. Very small $\eta$ (i.e., $\eta \leq 10^{-4}$) still doesn't change the model; however, larger $\eta$ (i.e., $10^{-4}$) produced good results. This implies that for any attack, our OTU approach could create a totally new encoder to be used in `MirrorCheck` by perturbing one or more of the mid-layers.

| Victim Model | Setting | One-Time-Use (OTU) ViT-B/32 Image Encoder | | | | |
| | | $\eta = 5 \cdot 10^{-6}$ | $\eta = 3 \cdot 10^{-4}$ | $\eta = 10^{-3}$ | $\eta = 10^{-2}$ | $\eta = 10^{-1}$ |
|---|---|---|---|---|---|---|
| UniDiffuser | Clean | 0.751 | 0.751 | 0.751 | 0.752 | **0.785** |
| | ADV-Transfer | 0.804 | 0.804 | 0.804 | 0.806 | 0.698 |
| BLIP | Clean | **0.728** | **0.728** | **0.728** | 0.730 | 0.775 |
| | ADV-Transfer | 0.478 | 0.478 | 0.478 | 0.481 | 0.587 |
| BLIP-2 | Clean | **0.750** | **0.750** | **0.750** | 0.752 | 0.791 |
| | ADV-Transfer | 0.499 | 0.499 | 0.499 | 0.501 | 0.615 |
| Img2Prompt | Clean | **0.695** | **0.695** | **0.695** | 0.697 | 0.767 |
| | ADV-Transfer | 0.478 | 0.478 | 0.478 | 0.480 | 0.589 |

Table 28: Finally, we investigate the model's performance when perturbing the last layers ($visual.ln\_post.weight$). **Key Takeaway**: This trend was the total opposite of what was observed when perturbing the first layer and when perturbing all weights. We observed good performances even when using large $\eta$ (up to $5 \cdot 10^{-1}$), which implies that there is a good range of encoders that can be created from a pretrained evaluator using our OTU approach. Furthermore, using this approach means that an attacker will find it difficult to create adversarial samples when carrying out an adaptive attack approach against our defense method.

| Victim Model | Setting | One-Time-Use (OTU) ViT-B/32 Image Encoder | | | | | |
| | | $\eta = 5 \cdot 10^{-6}$ | $\eta = 3 \cdot 10^{-4}$ | $\eta = 10^{-3}$ | $\eta = 10^{-2}$ | $\eta = 10^{-1}$ | $\eta = 5 \cdot 10^{-1}$ |
|---|---|---|---|---|---|---|---|
| UniDiffuser Bao et al. (2023b) | Clean | 0.751 | 0.751 | 0.751 | 0.751 | 0.748 | 0.723 |
| | ADV-Transfer | 0.804 | 0.804 | 0.804 | 0.804 | 0.801 | 0.781 |
| Blip Li et al. (2022) | Clean | **0.728** | **0.728** | **0.728** | **0.728** | 0.724 | **0.696** |
| | ADV-Transfer | 0.478 | 0.479 | 0.479 | 0.479 | 0.473 | 0.433 |
| Blip-2 Li et al. (2023b) | Clean | **0.750** | **0.750** | **0.750** | **0.750** | 0.746 | **0.720** |
| | ADV-Transfer | 0.499 | 0.499 | 0.499 | 0.498 | 0.492 | 0.450 |
| Img2Prompt Guo et al. (2023) | Clean | **0.695** | **0.695** | **0.695** | **0.695** | 0.689 | **0.656** |
| | ADV-Transfer | 0.478 | 0.478 | 0.478 | 0.478 | 0.472 | 0.429 |

## C.8 ADAPTIVE ATTACK

**Algorithm 1** Adaptive Attack to Bypass Defense Mechanisms (Figure 7)

1: **Input:** Original image $x_{in}$, target caption $t$
2: **Output:** Adversarial image $x_{adv}$
3: **Initialize:** $\delta \leftarrow 0$
4: **VLM Model:** $\mathcal{F}_\theta(x_{in}; p) \rightarrow t$      ▷ Victim model generates caption
5: **VLM Model Text Encoder:** $\hat{\mathcal{F}}_\theta(x_{in}) \rightarrow z$      ▷ Victim model text encoder generates embedding
6: **T2I Model Image Generator:** $\hat{G}_\psi(z) \rightarrow x_{gen}$      ▷ Generate image from text embedding
7: **Adapter Network Training:** Train adapter $\mathcal{A}$
8: **repeat**
9:      **Encode Input and Generated Images:**
10:      $x_{in} + \delta \rightarrow x_{adv}$
11:      $\hat{\mathcal{F}}_\theta(x_{adv}) \rightarrow z$
12:      $\mathcal{A}(z) \rightarrow z$
13:      $\hat{G}_\psi(z) \rightarrow x_{gen}$
14:      **for** $j = 1$ to $k$ **do**
15:          $\mathcal{I}_{\phi j, \xi}(x_{adv})$
16:          $\mathcal{I}_{\phi j, \xi}(x_{gen})$
17:      **end for**
18:      **Compute Loss:**
19:      loss $\leftarrow d(\tilde{\mathcal{I}}_\phi(x_{adv}), \tilde{\mathcal{I}}_\phi(G_\psi(t^*; \eta))) + \frac{1}{N}\sum_{j=1}^{N} d(\mathcal{I}_{\phi j, \xi}(x_{adv}), \mathcal{I}_{\phi j, \xi}(\hat{G}_\psi(\mathcal{A}(\hat{\mathcal{F}}_\theta(x_{adv}; p)); \eta)))$
20:      **Update** $\delta$:
21:      $\delta \leftarrow \delta - \gamma \cdot \nabla$loss
22: **until** Convergence
23: $x_{adv} \leftarrow x_{in} + \delta$
24: **return** $x_{adv}$

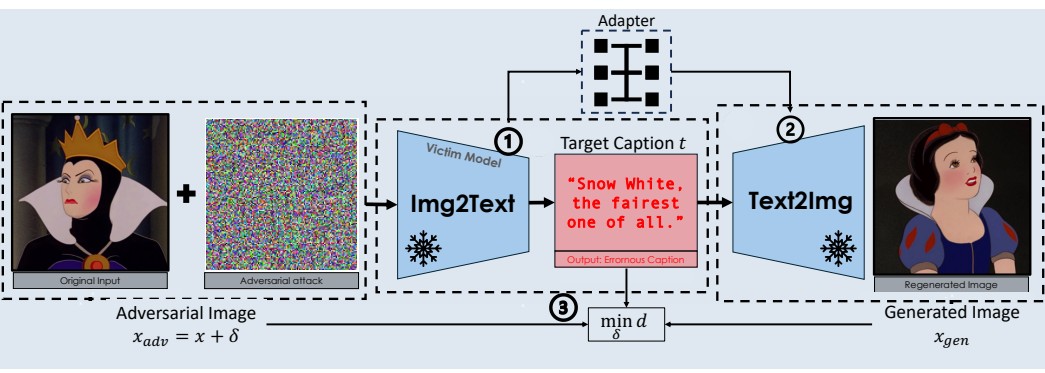

**Figure 7:** Illustration of the adaptive attack pipeline: (**1.**) Rather than use the discrete output of the Victim Model (I2T), the attacker seamlessly integrates the embedding layer for the text decoder (**2.**) with the decoding module of the generative model (T2I), using an *Adapter* for semantics alignment. The goal of the adapter is to (**3.**) craft adversarial images $x_{adv}$ such that its distance $d$ from target caption $t$ and generated images $x_{gen}$ is minimized.

We assessed the success of the adaptive attack relative to the standard attack, as the attacker's primary objective is to maximize the attack's effectiveness. To accomplish this, we computed the similarity between the target caption and the captions of the adversarial images generated by the adaptive attack. Subsequently, we compared these results with the similarity between the target caption and the captions of adversarial images generated by the standard attack (ADV-Transfer). As illustrated in Table 29, our findings indicate that the adaptive attack yields lower similarity, indicating a less successful attack.

**Table 29:** Text embedding similarity between the target captions and the captions produced by the victim model attacked using transfer attack (ADV-Transfer) and different adaptive settings.

| Victim Model | Setting | CLIP Image Encoder | | | | | |
| --- | --- | --- | --- | --- | --- | --- | --- |
| | | RN50 | RN101 | ViT-B/16 | ViT-B/32 | ViT-L/14 | Ensemble |
| UniDiffuser | ADV-Transfer | 0.76 | 0.71 | 0.74 | 0.77 | 0.68 | 0.73 |
| | Adaptive (ViT-B/32) | 0.59 | 0.61 | 0.60 | 0.64 | 0.53 | 0.60 |
| | Adaptive (RN50 + ViT-B/32) | 0.63 | 0.60 | 0.63 | 0.66 | 0.55 | 0.61 |
| | Adaptive (RN50 + ViT-B/32 + ViT-L/14) | 0.70 | 0.64 | 0.68 | 0.70 | 0.60 | 0.66 |
| | Adaptive (RN50 +ViT-B/16 + ViT-B/32 + ViT-L/14 ) | 0.69 | 0.64 | 0.68 | 0.71 | 0.62 | 0.67 |
| | Adaptive (RN50 + RN101 + ViT-B/16 + ViT-B/32 + ViT-L/14 ) | 0.63 | 0.64 | 0.66 | 0.67 | 0.58 | 0.64 |

# D  SOME VISUALIZATION

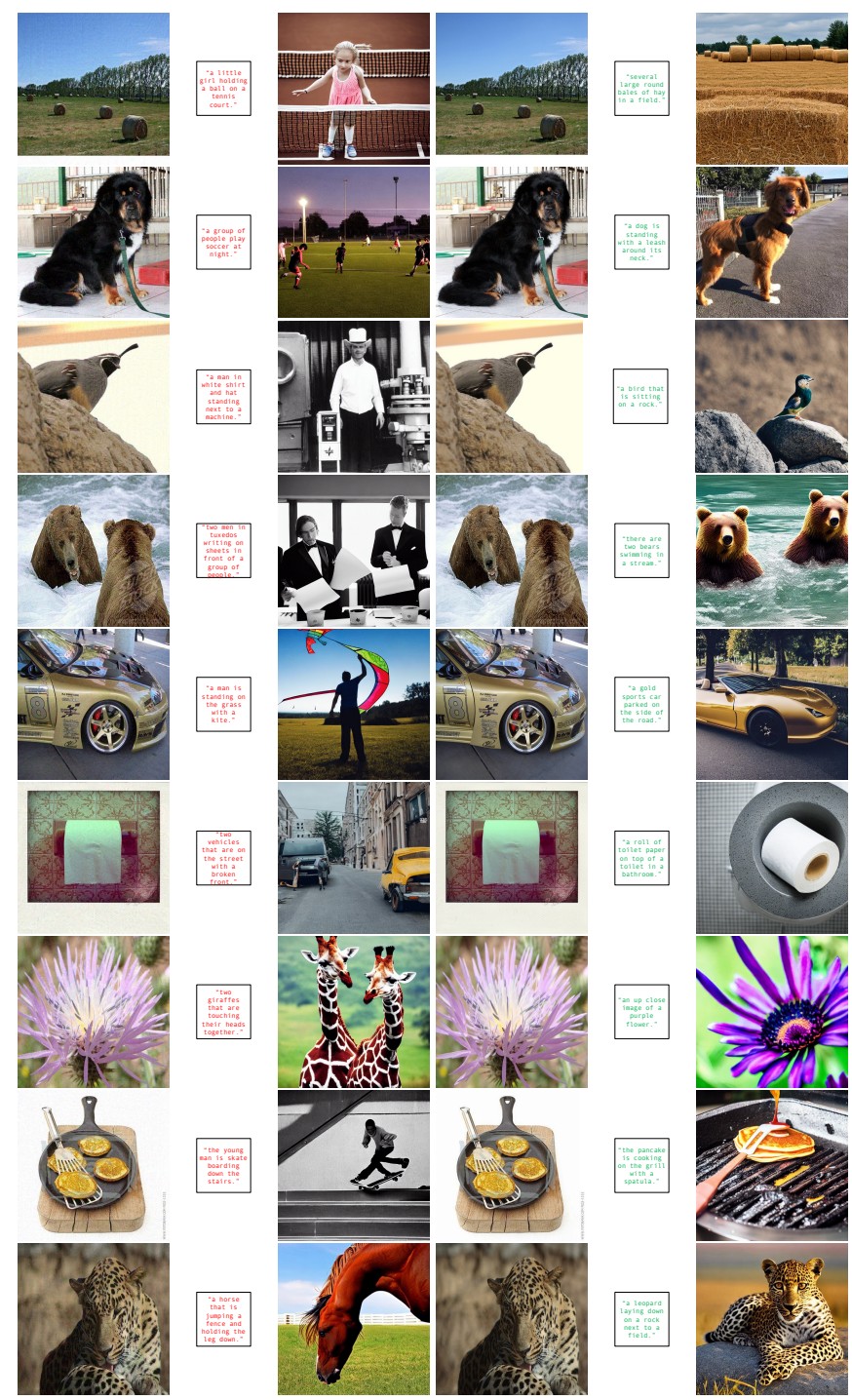

Figure 8: Visual results using BLIP (Victim Model) and Stable Diffusion (T2I Model).

