# OpenReview forum: "$\texttt{MirrorCheck}$ : Efficient Adversarial Defense for Vision-Language Models"
_ICLR.cc/2025/Conference — Submitted to ICLR 2025_

### Official Review · Reviewer_qvv1 · 2024-10-24

**Soundness:** 3
**Presentation:** 2
**Contribution:** 2
**Rating:** 5
**Confidence:** 4

**Summary:**

This paper focuses on detecting adversarial examples targeting Vision-Language Models (VLMs) by introducing a method called MirrorCheck. MirrorCheck uses an additional Text-to-Image (T2I) model and a CLIP image encoder to identify adversarial inputs. For any input to a VLM, the T2I model generates an image based on the VLM's output, and the CLIP image encoder extracts embeddings to compute a similarity score between the generated and original images. A significant difference in similarity indicates a potential adversarial example. The method is simple yet technically sound, and experiments with various VLMs, T2I models, and CLIP encoders confirm its effectiveness.

**Strengths:**

- Although the idea behind the method is simple, it is technically sound. The core concept is that adversarial examples tend to cause VLMs to generate text that deviates from the content of the original image. As a result, the T2I model's generated image will also deviate, leading to large differences in similarity scores between the generated and original images.
- While adversarial training is a popular defense method, it is costly, difficult to scale for large models, and faces trade-offs between robustness and accuracy. In contrast, detecting adversarial examples offers a cost-effective defense by rejecting malicious queries. MirrorCheck, the proposed detection method, is specifically designed for VLMs.
- The method is evaluated under a realistic black-box threat model, where the adversary can only query the victim model without access to gradient information, making it practical for real-world scenarios. Additionally, experiments were conducted using a white-box threat model to push the method to its limits. The evaluation setup and findings are well-appreciated.

**Weaknesses:**

- The core motivation behind MirrorCheck is that adversarial examples typically cause VLMs to generate content that deviates from the main image. However, adversaries could potentially manipulate the VLM to produce content that is aligned, or at least partially aligned, with the image. In such cases, MirrorCheck might fail to detect the adversarial example. For instance, consider Figure 2, where the adversary's target text is "A leopard laying down on a rock next to a field ...", followed by harmful or misleading texts. If the adversary uses a targeted attack to craft an adversarial example that closely aligns with the original image (except for the harmful text), how would the similarity scores behave? To what extent could this reduce MirrorCheck's detection effectiveness? This scenario raises concerns about MirrorCheck's robustness when handling adversarial attacks that maintain a degree of semantic consistency with the image. This type of targeted attack is examined in this work [1].

- The Average Similarity Scores in Table 1 alone do not fully capture MirrorCheck’s detection performance. While they show a general pattern of deviation between clean and adversarial examples, they do not clearly demonstrate whether MirrorCheck can reliably differentiate between them. In Table 3, the minimum similarity score for clean versus adversarial examples is on the same level, and a similar pattern is observed for the maximum scores, further obscuring the distinction. I recommend that the authors plot the distribution of similarity scores to better illustrate the detection performance.

- The detection accuracy, TPR, and FPR presented in Tables 2 and 4 are influenced by the ratio of adversarial and clean examples. However, this ratio is not explicitly mentioned in the paper. In real-world scenarios, adversarial queries are typically rare. I suggest conducting an ablation study that examines detection performance under varying ratios of adversarial examples. Starting with a balanced 50%-50% to 99.9%-0.1%

- Line 371 stated that the threshold $\tau$ is based on optimized performance. In practical settings, it is very unlikely that the optimal hyperparameters can be found. The similarity scores could vary depending on the validation set, attack used, and CLIP image encoder. I suggest also including the area under the ROC curve and Precision-Recall curve.

- Following the setting in the adaptive attack study, line 485, the adversarial examples can still be detected by other encoders. This could be the result of the adversarial example not transferable to other encoders. What if the adversary improves the transferability by using an ensemble of other encoders?

- There are relevant detection baselines [2-4] and related adversarial attacks [1, 5, 6] that are not examined in the paper.

- Line 098 claims that MirrorCheck does not require training. This could be misleading. T2I models and CLIP encoders are part of the defense that does require training.

- Lines 177 - 182 suggest that existing defenses do not consider multi-modal settings. However, several works [7-9] have proposed adversarial training for the multi-modal setting.

---

[1] Schlarmann, C., & Hein, M. (2023). On the adversarial robustness of multi-modal foundation models. In Proceedings of the IEEE/CVF International Conference on Computer Vision(pp. 3677-3685).\
[2] Lee, K., Lee, K., Lee, H., & Shin, J. (2018). A simple unified framework for detecting out-of-distribution samples and adversarial attacks. Advances in neural information processing systems, 31.\
[3] Ma, X., Li, B., Wang, Y., Erfani, S. M., Wijewickrema, S., Schoenebeck, G., ... & Bailey, J. (2018). Characterizing Adversarial Subspaces Using Local Intrinsic Dimensionality. In International Conference on Learning Representations.\
[4] Cohen, G., Sapiro, G., & Giryes, R. (2020). Detecting adversarial samples using influence functions and nearest neighbors. In Proceedings of the IEEE/CVF conference on computer vision and pattern recognition (pp. 14453-14462).\
[5] Bailey, L., Ong, E., Russell, S., & Emmons, S. (2023). Image hijacks: Adversarial images can control generative models at runtime. arXiv preprint arXiv:2309.00236.\
[6] Dong, Y., Chen, H., Chen, J., Fang, Z., Yang, X., Zhang, Y., ... & Zhu, J. (2023). How Robust is Google's Bard to Adversarial Image Attacks?. arXiv preprint arXiv:2309.11751.\
[7] Mao, C., Geng, S., Yang, J., Wang, X., & Vondrick, C. (2023). Understanding Zero-shot Adversarial Robustness for Large-Scale Models. In The Eleventh International Conference on Learning Representations.\
[8] Schlarmann, C., Singh, N. D., Croce, F., & Hein, M. (2024). Robust CLIP: Unsupervised Adversarial Fine-Tuning of Vision Embeddings for Robust Large Vision-Language Models. In Forty-first International Conference on Machine Learning.\
[9] Wang, S., Zhang, J., Yuan, Z., & Shan, S. (2024). Pre-trained model guided fine-tuning for zero-shot adversarial robustness. In Proceedings of the IEEE/CVF Conference on Computer Vision and Pattern Recognition (pp. 24502-24511).

**Questions:**

- Line 158-160, the certified defense methods can guarantee robustness. What does it mean by the struggle to generalize to novel attack strategies?
- Line 253, experiments are conducted with 100 or 1000 randomly selected images; why not use the full dataset for evaluation? And which experiments are using 100 and which are using 1000?
- Line 258-260, I believe the PGD requires a gradient for optimization of the attack vector; how is this used for 100 queries and 8-step PGD with an estimated gradient?

---

> ### Author Response · Authors · 2024-11-22
>
> ```
> The Average Similarity Scores in Table 1 alone do not fully capture MirrorCheck’s detection performance. While they show a general pattern of deviation between clean and adversarial examples, they do not clearly demonstrate whether MirrorCheck can reliably differentiate between them. I recommend that the authors plot the distribution of similarity scores to better illustrate the detection performance.
> ```
> - To address this, we have included graphs that plot the distribution of similarity scores for both clean and adversarial examples across four victim models (see Figure 2 in the link below).
> - Link: https://anonymous.4open.science/r/MC-6200/MirrorCheck_IMG.pdf
>
> ```
> The detection accuracy, TPR, and FPR presented in Tables 2 and 4 are influenced by the ratio of adversarial and clean examples. However, this ratio is not explicitly mentioned in the paper. In real-world scenarios, adversarial queries are typically rare. I suggest conducting an ablation study that examines detection performance under varying ratios of adversarial examples.
> ```
> - In **Figure 3 in the link provided above**, we included a confusion matrix that provides results across different ratios of clean examples. This offers insights into how the detection performance varies with varying proportions of clean and adversarial queries.
>
> ```
> Following the setting in the adaptive attack study, line 485, the adversarial examples can still be detected by other encoders. This could be the result of the adversarial example not transferable to other encoders. What if the adversary improves the transferability by using an ensemble of other encoders?
> ```
> - We would like to clarify that this scenario is already addressed in our adaptive attack experiments, both in the main paper and detailed further in the Appendix. Specifically, we considered adversaries employing ensembles of encoders to enhance transferability and evaluated MirrorCheck's robustness in these challenging settings. The results demonstrated that the use of diverse or perturbed encoders in our detection pipeline (e.g., one-time-use encoders) effectively counters such attacks, as adversarial examples generated for one set of encoders often fail to transfer to others. Moreover, increasing the diversity of encoders in the defense ensemble
>
> ```
> There are relevant detection baselines [2-4] and related adversarial attacks [1, 5, 6] that are not examined in the paper.
> ```
> - We thank the reviewer for highlighting these relevant detection baselines and adversarial attacks. However, [2-4] focus on unimodal detection or rely on DNN-specific properties (e.g., spatial features or internal architecture access), directly comparing with our multi-modal, black-box, and zero-shot VLM-based approach infeasible without significant adaptation.  While [5] introduces a novel attack methodology, we were unable to reproduce the results due to technical constraints in implementing their attack pipeline.
>
> ```
> Line 098 claims that MirrorCheck does not require training. This could be misleading. T2I models and CLIP encoders are part of the defense that does require training.
> ```
> - We appreciate the reviewer’s observation regarding the training requirements of T2I models and CLIP encoders. To clarify, when we state in line 098 that MirrorCheck "does not require training," we are referring specifically to the fact that our defense mechanism itself does not require any additional training or fine-tuning. Instead, it leverages pre-trained components such as T2I models and image encoders, which are widely available and not specific to our framework.
>
> ```
> Lines 177 - 182 suggest that existing defenses do not consider multi-modal settings. However, several works [7-9] have proposed adversarial training for the multi-modal setting.
> ```
> - We thank the reviewer for pointing out references [7-9] and highlighting prior work on adversarial training in multi-modal settings. Upon review, we acknowledge that adversarial training has been explored for specific multi-modal tasks. However, our intention was to emphasize the lack of tailored detection strategies explicitly designed for Vision-Language Models (VLMs) that are agnostic to the model architecture and attack methodology. While adversarial training methods are indeed valuable, they often require modifications to the model and are computationally expensive, particularly for large-scale VLMs. In contrast, MirrorCheck focuses on a detection-based approach that operates in a zero-shot setting, leveraging Text-to-Image (T2I) models and pretrained image encoders without altering the underlying VLM or requiring adversarial training. To clarify this distinction, we will revise the corresponding lines in the manuscript to explicitly acknowledge adversarial training methods in multi-modal contexts while differentiating our work as a detection framework uniquely tailored to VLMs.

---

> > ### Author Response · Authors · 2024-11-22
> >
> > ```
> > The core motivation behind MirrorCheck is that adversarial examples typically cause VLMs to generate content that deviates from the main image. However, adversaries could potentially manipulate the VLM to produce content that is aligned, or at least partially aligned, with the image. In such cases, MirrorCheck might fail to detect the adversarial example. For instance, consider Figure 2, where the adversary's target text is "A leopard laying down on a rock next to a field ...", followed by harmful or misleading texts. If the adversary uses a targeted attack to craft an adversarial example that closely aligns with the original image (except for the harmful text), how would the similarity scores behave? To what extent could this reduce MirrorCheck's detection effectiveness? This scenario raises concerns about MirrorCheck's robustness when handling adversarial attacks that maintain a degree of semantic consistency with the image. This type of targeted attack is examined in this work [1].
> >
> > ```
> > - We acknowledge the importance of the scenario the reviewer raised. While [1] examines this type of targeted attack, the corresponding code is not currently available. However, we are actively working on implementing this attack and will update the results as soon as they are available. Additionally, we note that similar trends could be observed with Attack-Bard [6] which we evaluated and reported in (Answer 2) to the first reviewer. Thank you for highlighting this valuable perspective.

---

> ### Comment · Reviewer_qvv1 · 2024-11-23
> **Thanks for the response**
>
> Thanks for the response, which addressed my concerns about the distribution of similarity scores and clean sample ratio. I'm looking forward to the author's clarification on the following, as well as questions in the initial review.
>
> > unimodal detection baselines.
>
> I believe it is a fair comparison for unimodal detection in image space. For multiple modal queries, these detections could use the images for detection and disregard other modalities. For MirrorCheck, as it is specially designed for multi-modal models, it should present a clear advantage. Could authors clarify what makes unimodal detection not applicable to the detection?
>
> > MirrorCheck "does not require training.
>
> Relying on pre-trained models doesn't necessarily mean training is not required since these models are part of the defense. If relying on these pre-trained models, attackers could generate adaptive attacks using these pre-trained models. Defenders might want to use customized T2I models (require training). I'm happy to set this aside while the author works on the rest of the questions.
>
> > Adversarial training is computationally expensive.
>
> I would like to point out that these are finetuning, not pre-training. It is true typical adversarial training is computationally expensive. But finetuning is not very expensive. It is a suggestion for more comprehensive related work discussions, not a major weakness, and I'm happy to mark it as addressed.
>
> > While [1] examines this type of targeted attack, the corresponding code is not currently available.
>
> The code is available here: [https://github.com/chs20/robustvlm](https://github.com/chs20/robustvlm)
>
> This is one of my main concerns is that the adversary adaptively evades detection.
>
> I would like to maintain my rating for now.

---

> ### Author Response · Authors · 2024-11-24
>
> We appreciate the reviewer for the suggestions and comments, as well as providing the resource to the code for the RobustVLM attacks. We have now carried out more experiments to further show the efficiency and effectiveness of our method.
>
> ```
> Line 158-160, the certified defense methods can guarantee robustness. What does it mean by the struggle to generalize to novel attack strategies?
> ```
>
> **Response**: We were referring to adversarial training, we will correct this in the main manuscript.
>
> ```
> Line 253, experiments are conducted with 100 or 1000 randomly selected images; why not use the full dataset for evaluation? And which experiments are using 100 and which are using 1000?
> ```
>
> **Response**: Tables 1 and 2 in the main manuscript are results based on 100 samples. We validated the results by running on 1000 samples (Table 3 and 4). Table 3 and 4 proves that we could get generalizable results using just 100 images.
>
> ```
> Line 258-260, I believe the PGD requires a gradient for optimization of the attack vector; how is this used for 100 queries and 8-step PGD with an estimated gradient?
> ```
>
> **Response**: We followed the query-based attacking strategy from "On Evaluating Adversarial Robustness of Large Vision-Language Models." This approach estimates gradients using the Random Gradient-Free (RGF) method, which approximates the gradient by sampling random directions and measuring changes in similarity between the victim model's output and the target response. For 100 queries and 8-step PGD, these approximated gradients are iteratively used to refine the adversarial image within the allowed perturbation. They also consider the adversarial examples generated by transfer-based methods to be an initialization.
>
> ```
> unimodal detection baselines.
> ```
>
> **Response**: While valuable in unimodal settings, these methods face limitations when applied to Vision-Language Models (VLMs). For instance, "A Simple Unified Framework for Detecting Out-of-Distribution Samples and Adversarial Attacks" relies on softmax confidence scores, which are not directly applicable to VLMs' multi-modal outputs without significant adaptation. "Characterizing Adversarial Subspaces Using Local Intrinsic Dimensionality (LID)" requires intermediate activations and focuses on spatial properties of unimodal models, which do not align with MirrorCheck's approach of detecting semantic deviations between input and regenerated images. Similarly, "Detecting Adversarial Samples Using Influence Functions and Nearest Neighbors (NNIF)" depends on internal architecture access and training data, making it incompatible with black-box VLMs. Unlike these methods, MirrorCheck is specifically designed to detect adversarial perturbations by evaluating semantic consistency between input and reconstructed images, making direct comparisons challenging. We will revise the manuscript to include this discussion.
>
> ```
> While [1] examines this type of targeted attack, the corresponding code is not currently available. The code is available here: https://github.com/chs20/robustvlm. This is one of my main concerns is that the adversary adaptively evades detection.
> ```
>
> **Response**: We thank the reviewer for providing the link to the code. Following the same attack setup described in the repository, we conducted experiments on LLaVA and OpenFlamingo for the VQA task, using 500 images from each attacked model. Below are the detection results for MirrorCheck, based on **thresholds calculated for a different model (Img2Prompt) from Table 1 in the main manuscript**:
>
> | Attack                                      | RN50  | RN101 | ViT-B/16 | ViT-B/32 | ViT-L/14 | Ensemble |
> |--------------------------------------------|-------|-------|----------|----------|----------|----------|
> | LLaVA  | 0.734 | 0.779 | 0.755    | 0.761    | 0.728    | 0.761    |
> | OpenFlamingo  | 0.754 | 0.792 | 0.788    | 0.787    | 0.751    |          |
>
> We observed consistent detection performance across these semantic attacks, with detection accuracies exceeding 73\% in all cases. These results highlight the robustness of MirrorCheck against the semantic attack scenario suggested.
>
> Furthermore, we also experimented with **tuning the thresholds** specifically for this attack setup. Interestingly, the performance was not significantly affected, demonstrating that MirrorCheck is less sensitive to the threshold value, further underscoring the method’s robustness. For reference, the results with tuned thresholds are shown below:
>
> | Attack                          | RN50  | RN101 | ViT-B/16 | ViT-B/32 | ViT-L/14 |
> |---------------------------------|-------|-------|----------|----------|----------|
> | LLaVA    | 0.761 | 0.780 | 0.772    | 0.774    | 0.738    |
> | OpenFlamingo | 0.755 | 0.791 | 0.796    | 0.790    | 0.760    |
>
> Thank you for raising this critical concern! We hope these results provide clarity and address the points raised in your feedback.

---

> > ### Comment · Reviewer_qvv1 · 2024-11-28
> >
> > I appreciate the new results and clarifications. The findings on targeted attacks suggest that detection performance could be affected. Compared to the 90% or higher detection accuracy achieved with untargeted attacks, the accuracy drops to approximately 70%, indicating the potential for adaptive attacks to bypass detection mechanisms.
> >
> > Thank you for the effort in providing these new results. I have raised my rating to 5. While I will not advocate for rejection, I would need further convincing from other reviewers to increase my score further at this stage. I recommend including more analysis on adaptive attacks in the next revision.

---

> > > ### Author Response · Authors · 2024-11-28
> > >
> > > Thank you for your feedback. We appreciate your acknowledgement of our results and clarifications. We will include a more detailed analysis of adaptive attacks in the next revision to address this important aspect.

---

> ### Author Response · Authors · 2024-11-27
>
> ```
> Relying on pre-trained models doesn't necessarily mean training is not required since these models are part of the defense. If relying on these pre-trained models, attackers could generate adaptive attacks using these pre-trained models. Defenders might want to use customized T2I models (require training).
> ```
>
> Thank you for your insightful comment.
>
> We would like to address your concern by clarifying that our adaptive attack study already considers scenarios where attackers have full knowledge of our defense pipeline, including the use of pre-trained models. While defenders can choose to customize and train their Text-to-Image (T2I) models, our results indicate that this is not strictly necessary.
>
> In cases where the attacker is aware of the defense mechanism, our proposed One-Time-Use (OTU) approach allows the defender to use different variations of the same model dynamically, making it more challenging for attackers to generate successful adaptive attacks. Furthermore, our strongest defense pipeline, which uses an ensemble of models, has proven to be highly robust, significantly reducing the likelihood of successful attacks.
>
> Importantly, defenders do not need to retrain models; instead, simple customizations, such as injecting noise (OTU) into specific layers of the pre-trained models, can enhance robustness. This eliminates the need for costly retraining while maintaining the flexibility to counter adaptive threats.
>
> We hope this clarifies your concerns and demonstrates the effectiveness of our approach. Thank you for dedicating your time reviewing our paper and providing valuable feedback.
>
> If you feel that we have adequately addressed your concerns, we would be grateful if you would consider reassessing your rating.
>
> We would be happy to clarify or elaborate on any part of our paper while the discussion period is still open.
>
> Thank you!

---

### Official Review · Reviewer_RQhj · 2024-10-30

**Soundness:** 2
**Presentation:** 3
**Contribution:** 3
**Rating:** 5
**Confidence:** 5

**Summary:**

This paper presents a simple adversarial sample detection method called MirrorChecker, which detects adversarial samples by comparing the embedding similarity between reconstructed adversarial samples and clean samples. Specifically, given an adversarial sample, the method first uses a pre-trained multimodal model to generate a caption for the image and then employs a diffusion model to reconstruct the image based on that caption. Finally, it assesses the similarity between the reconstructed image and the clean image to detect malicious adversarial samples. The paper validates the effectiveness of the proposed defense method across multiple models.

**Strengths:**

- The writing is clear.
- Detecting adversarial samples in multimodal models is an interesting research topic.

**Weaknesses:**

- **Unclear Threat Model:** In section 3.1, the authors focus on targeted attacks (lines 199-200), but this assumption is unclear. It is recommended that the authors address a broader range of attack scenarios, including patch attacks and untargeted attacks. Specifically, how might untargeted attacks affect the performance of the proposed detection method?

- **Lack of Defender Capability:** The paper does not provide sufficient detail about the defender's capabilities and the practical feasibility of the proposed defense. A key concern is how the defender obtains the clean samples needed for comparison. In other words, if clean images are not required, how does the defense work? The authors should clarify how this affects the real-world applicability of their approach.

- **No Support for Theoretical Analyses:** The authors claim to provide theoretical analyses in the abstract (line 22); however, a review of the paper reveals no such analyses. If these analyses are not present, the authors should explain why this claim is made.

- **Insufficient Robustness Against Adaptive Attacks:** The robustness of the proposed defense method against adaptive attacks needs further analysis. For example, if attackers are aware of this defense mechanism, they may modify the adversarial images to minimize the distance to the clean images. Would this enable them to evade the proposed detection? The authors should consider stronger adaptive attacks and provide corresponding defense results.

- **Defense Results on Out-of-Distribution Data:** It is unclear how the proposed method detects adversarial samples when applied to out-of-distribution (OOD) data, particularly across multiple domains. The authors should provide either experimental results or a theoretical discussion on the performance of their method when dealing with OOD samples from various domains.

- **Lack of Comparison with Advanced Defense Baselines:** A comparison with advanced baselines, such as DiffPure [1], would strengthen the paper. DiffPure uses a diffusion model to remove adversarial noise and detects adversarial samples by assessing the prediction differences between cleaned samples and the original adversarial samples. The detection process of DiffPure seems more straightforward and efficient than the proposed method. The authors should provide a quantitative comparison on a common dataset, focusing on both detection accuracy and computational efficiency. This would help highlight the strengths of their method over existing baselines.

[1] Diffusion models for adversarial purification, ICML, 2022

**Questions:**

see weakness.

---

> ### Author Response · Authors · 2024-11-22
>
> ```
> Unclear Threat Model: In section 3.1, the authors focus on targeted attacks (lines 199-200), but this assumption is unclear. It is recommended that the authors address a broader range of attack scenarios, including patch attacks and untargeted attacks. Specifically, how might untargeted attacks affect the performance of the proposed detection method?
> ```
> - While the primary focus of our experiments was on targeted attacks, we also conducted additional experiments to evaluate the performance of MirrorCheck against untargeted attacks. These results are presented in Appendix C4, where we demonstrate that our method is effective across both attack types. Moreover, we have included further experiments to assess MirrorCheck’s robustness against Attack-Bard, an untargeted attack designed for VLMs [1] (https://arxiv.org/pdf/2309.11751). The results are detailed in the table below. This comprehensive evaluation confirms the versatility and reliability of our approach under various adversarial scenarios.
>
> - Reference: [1] Dong, Yinpeng, et al. "How Robust is Google's Bard to Adversarial Image Attacks?." arXiv preprint arXiv:2309.11751 (2023).
> | Victim Model | Task | Setting      | RN50   | RN101  | ViT-B/16 | ViT-B/32 | ViT-L/14 | Ensemble |
> |--------------|------|--------------|--------|--------|----------|----------|----------|----------|
> | BLIP-2       | ID   | Attack-Bard  | 0.850  | 0.835  | 0.890    | 0.975    | 0.905    | 0.891    |
>
> ```
> Lack of Defender Capability: The paper does not provide sufficient detail about the defender's capabilities and the practical feasibility of the proposed defense. A key concern is how the defender obtains the clean samples needed for comparison. In other words, if clean images are not required, how does the defense work? The authors should clarify how this affects the real-world applicability of their approach.
> ```
> - We would like to clarify that MirrorCheck does not rely on access to clean images for its operation. Instead, our defense mechanism functions by comparing the input image—whether clean or adversarial—with a regenerated image produced from the Vision-Language Model's (VLM) generated caption using a Text-to-Image (T2I) model. This regenerated image acts as a proxy for a "clean" sample, capturing the semantic consistency of the input image and its corresponding caption. If the input image has been adversarially perturbed, the semantic consistency between the input and regenerated images is disrupted, resulting in a significant deviation in their embeddings. This deviation is used as the basis for detecting adversarial samples, based on a threshold $\tau$. Defenders can establish a suitable threshold $\tau$ for detection by using a representative dataset to calibrate our proposed defense mechanism. This approach ensures that MirrorCheck remains effective and applicable in real-world scenarios without the need for explicitly clean images.
>
> ```
> No Support for Theoretical Analyses: The authors claim to provide theoretical analyses in the abstract (line 22); however, a review of the paper reveals no such analyses. If these analyses are not present, the authors should explain why this claim is made.
> ```
> - We thank the reviewer for raising this concern. We would like to clarify that the theoretical foundation of MirrorCheck is indeed discussed in the Appendix under the section "MIRRORCHECK AS AN AUTOENCODER". In this section, we draw an analogy between MirrorCheck and traditional autoencoder-based anomaly detection methods. We acknowledge that this discussion is currently limited to the Appendix, and its placement may not highlight its significance. To address this, we will mention the theoretical foundation in the main text and point readers explicitly to the Appendix for details and clarify in the abstract that theoretical analyses refer to this conceptual analogy and its implications for robustness against adversarial attacks. We thank the reviewer for bringing this to our attention, and we will revise the manuscript to improve clarity and emphasis on this theoretical aspect.

---

> > ### Author Response · Authors · 2024-11-22
> >
> > ```
> > Insufficient Robustness Against Adaptive Attacks: The robustness of the proposed defense method against adaptive attacks needs further analysis. For example, if attackers are aware of this defense mechanism, they may modify the adversarial images to minimize the distance to the clean images. Would this enable them to evade the proposed detection? The authors should consider stronger adaptive attacks and provide corresponding defense results.
> > ```
> > - We would like to emphasize that we have already evaluated MirrorCheck under several adaptive attack scenarios, including those where attackers attempt to evade detection by minimizing the similarity gap between adversarial images and their regenerated counterparts. These results are detailed in the **main text (lines 447-488 and Table 5)** and further elaborated in **Appendix C6**. We considered scenarios where attackers use the same encoder or a subset of encoders used in the detection pipeline to maximize transferability and similarity with regenerated images. Our experiments also included attackers leveraging Expectation Over Transformation (EOT) techniques and ensemble-based methods to enhance the transferability of adversarial examples. **By using multiple, diverse pretrained encoders**, we make it challenging for attackers to optimize adversarial examples that transfer across all encoders. Perturbing the encoder weights ensures that the specific detection pipeline is resistant to adaptive attacks targeting static encoders. MirrorCheck fundamentally relies on the semantic inconsistency introduced by adversarial perturbations, which remains a challenging target for attackers attempting to mimic the original image’s semantics. The dynamic nature of OTU encoders and the use of diverse ensembles significantly complicate an attacker's optimization process. We thank the reviewer for this suggestion, as it provides an opportunity to better emphasize the adaptive robustness of our method.
> >
> > ```
> > Defense Results on Out-of-Distribution Data: It is unclear how the proposed method detects adversarial samples when applied to out-of-distribution (OOD) data, particularly across multiple domains. The authors should provide either experimental results or a theoretical discussion on the performance of their method when dealing with OOD samples from various domains.
> > ```
> > - We emphasize in our manuscript that MirrorCheck is designed to be model-agnostic, allowing flexibility in selecting pretrained Text-to-Image (T2I) models and image encoders for computing similarity scores. This adaptability means that, in practical applications, it is crucial to deploy models that are well-suited to the specific domain or use case of the victim model to achieve optimal performance. While our primary focus is not on OOD data, our approach inherently allows customization to improve effectiveness in different domains.
> >
> > ```
> > Lack of Comparison with Advanced Defense Baselines: A comparison with advanced baselines, such as **DiffPure**, would strengthen the paper. DiffPure uses a diffusion model to remove adversarial noise and detects adversarial samples by assessing the prediction differences between cleaned samples and the original adversarial samples. The detection process of DiffPure seems more straightforward and efficient than the proposed method. The authors should provide a quantitative comparison on a common dataset, focusing on both detection accuracy and computational efficiency. This would help highlight the strengths of their method over existing baselines.
> > ```
> > - Thank you for suggesting a comparison with DiffPure. We would like to clarify that DiffPure is primarily designed for purification rather than explicit adversarial detection. While it uses a diffusion model to remove adversarial noise, DiffPure does not include a dedicated detection module. Furthermore, DiffPure’s effectiveness relies on retraining the model to adapt to each new type of attack, which can be computationally expensive and less efficient (as stated in the concluding part of the paper). In contrast, our method does not require any retraining or additional training steps, making it inherently more efficient. Additionally, we have the potential to further optimize our approach by parallelizing certain processes and reducing the timesteps used for image generation. However, a direct performance comparison with DiffPure may not be entirely appropriate, as our method is focused specifically on adversarial detection, whereas DiffPure is focused on purification.

---

> > > ### Comment · Reviewer_RQhj · 2024-11-23
> > >
> > > Thanks for the response, which addressed some of my concerns. Although the author clarified that the proposed method does not rely on clean samples but instead determines semantic consistency by reconstructing the before-and-after image semantics, there are still some issues with the proposed approach:
> > >
> > > - Semantic adversarial sample interference: Attackers can subtly manipulate the semantics to ensure semantic consistency between the before-and-after images, thereby evading detection.
> > > - Presence of OOD (out-of-distribution) data or other natural adversarial samples: This could challenge the accuracy of the detection and the setting of thresholds.
> > > - DiffPure can serve as a detection method:  Although the design motivations differ, DiffPure seems more efficient than the method proposed in this paper. The author should not overlook this fact.
> > >
> > > Based on the above points, I have appropriately increased the score to 5.

---

> > > > ### Comment · Reviewer_RQhj · 2024-11-26
> > > >
> > > > Thank you for your response.  My main concern is that the proposed defense may be vulnerable to adaptive attacks. This is because the text2img step relies on a pre-trained model, which seems to be an easily exploitable step for an attacker. For instance, in the case of adaptive attacks, the attacker could preemptively add regularization to the text2image encoder, thereby reducing the semantic distance between the original and reconstructed outputs while maintaining the original attack objective.
> > > >
> > > > Based on this consideration, I will keep the current score unchanged.

---

> ### Author Response · Authors · 2024-11-24
>
> We sincerely thank the reviewer for engaging in the discussion and providing valuable insights that have helped strengthen our contribution.
>
> ```
> DiffPure can serve as a detection method: Although the design motivations differ, DiffPure seems more efficient than the method proposed in this paper. The author should not overlook this fact.
> ```
>
> **Response**: We thank the reviewer for the suggestions regarding DiffPure and appreciate the opportunity to clarify our comparisons and findings. We have now conducted experiments with DiffPure (results shown in the table below) and demonstrated that our method achieves superior detection performance. Below, we outline the key differences between DiffPure and MirrorCheck, along with the results of our comparative analysis:
>
> 1.  **Detection vs. Purification**: DiffPure was originally designed for purification, not detection. To use DiffPure as a detection pipeline in our experiments, we passed each image through its purification pipeline and compared the embedding of the purified image to that of the original image. As seen in the table below, MirrorCheck consistently outperforms DiffPure in terms of detection accuracy across various model architectures.
>
> 2.  **Efficiency**: In our experiments, DiffPure required approximately 8 seconds per image on an RTX A6000 GPU, compared to MirrorCheck's 15 seconds per image at 50 timesteps. However, we optimized MirrorCheck by reducing the number of timesteps to 10, enabling it to process 100 images in 2 minutes (1.2 seconds per image) while maintaining a higher detection accuracy than DiffPure. This demonstrates MirrorCheck's potential for further optimization to significantly reduce processing times without substantial performance degradation.
>
> 3.  **Model-Agnostic Nature**: MirrorCheck is model-agnostic, meaning it is not tied to specific architectures or datasets. This flexibility makes it more difficult for attackers to create adaptive attacks against our method. Furthermore, the adaptable nature of MirrorCheck has been leveraged in other research (details withheld for blind review) to defend against jailbreaking threats. Additionally, optimizing MirrorCheck for faster performance is straightforward, as reducing the number of timesteps in the T2I model directly reduces processing time while maintaining competitive detection accuracy.
>
> | Defense            | RN50 | RN101 | ViT-B/16 | ViT-B/32 | ViT-L/14 | Ensemble |
> |--------------------|------|-------|----------|----------|----------|----------|
> | DiffPure           | 0.65 | 0.61  | 0.64     | 0.62     | 0.76     | 0.65     |
> | MirrorCheck        | 0.89 | 0.93  | 0.84     | 0.90     | 0.80     | 0.90     |
> | MirrorCheck (t=10) | 0.87 | 0.85  | 0.83     | 0.89     | 0.78     | 0.84     |
>
> While DiffPure and MirrorCheck have different design motivations (purification vs. detection), our results show that MirrorCheck offers significant advantages in terms of detection performance and adaptability, while optimizations could boost efficiency. We will update the manuscript accordingly and include the DiffPure results.
>
> ```
> Presence of OOD (out-of-distribution) data or other natural adversarial samples: This could challenge the accuracy of the detection and the setting of thresholds.
> ```
> **Response**: Thank you for raising this important point. We acknowledge that the presence of OOD data can pose challenges to the accuracy of detection and the setting of thresholds. Our current approach assumes that the input data generally aligns with the domain of the pretrained generative model and image encoder. To address this, we recommend calibrating the detection threshold $\tau$ using a representative dataset that includes a mix of in-distribution and plausible OOD samples during the setup phase. We will also recommend employing adaptive thresholding techniques or integrating a separate OOD detection module, which could further enhance the robustness of MirrorCheck in such scenarios. While our current focus has not been explicitly on handling OOD data, we believe this is an interesting direction for future work.

---

> ### Author Response · Authors · 2024-11-24
>
> ```
> Semantic adversarial sample interference: Attackers can subtly manipulate the semantics to ensure semantic consistency between the before-and-after images, thereby evading detection.
> ```
> The scenario described aligns with the objectives of the AttackVLM framework [1], which we used to generate adversarial attacks in our experiments. In these attacks, image semantics are preserved to challenge detection systems. However, achieving a much higher semantic consistency while ensuring a successful attack often requires reducing the attack strength, which can limit the attack's effectiveness.
>
> For cases where the attacks are successful while maintaining semantic consistency, our experimental results still demonstrate that MirrorCheck remains robust in detecting such samples. These findings are detailed in our manuscript as well as Figure 1 in [2], showcasing the effectiveness of our approach even under challenging scenarios where semantics are subtly manipulated.
>
> Additionally, **Reviewer qvv1** raised a similar concern and suggested an attack implementation for such scenarios [3]. We followed their suggestion and conducted experiments using the same attack setup, as detailed in our response. Please refer to the updated results provided in our last [response](https://openreview.net/forum?id=p4jCBTDvdu&noteId=c51QaVGiJa), where we demonstrate that MirrorCheck remains robust against these semantic attacks.
>
> [1] Zhao, Y., Pang, T., Du, C., Yang, X., Li, C., Cheung, N. M. M., \& Lin, M. (2024). On evaluating adversarial robustness of large vision-language models. Advances in Neural Information Processing Systems, 36.
>
> [2] https://anonymous.4open.science/r/MC-6200/MirrorCheck\_IMG.pdf
>
> [3] Schlarmann, C., & Hein, M. (2023). On the adversarial robustness of multi-modal foundation models. In Proceedings of the IEEE/CVF International Conference on Computer Vision(pp. 3677-3685).

---

> > ### Author Response · Authors · 2024-11-25
> > **Grateful for your suggestions and awaiting further discussion**
> >
> > Dear Reviewer RQhj,
> >
> > Thank you for dedicating your time reviewing our paper and providing valuable feedback.
> >
> > We have thoughtfully addressed each of your comments, offering detailed responses to clarify and resolve the concerns you raised. We hope our explanations have provided a clearer perspective on our work and its contributions.
> >
> > If you feel that we have adequately addressed your concerns, we would be grateful if you would consider reassessing your rating.
> >
> > We would be happy to clarify or elaborate on any part of our paper while the discussion period is still open.
> >
> > Thank you!

---

> ### Author Response · Authors · 2024-11-26
>
> Thank you for your concern and suggestion regarding adaptive attacks. We tried incorporating the similarity between the adversarial (adv) and generated (gen) images into the optimization of the original attack as you proposed. Here’s what we found: By adding the similarity measure between the adv and gen images to the original attack using the same encoder as the attack, we observed that the attack was effective, and the similarity was indeed high between the adv and gen images. However, the adversarial image was still detected by other encoders. When we employed a different encoder for optimizing the similarity between the adv and gen image, the similarity score decreased in the original attack. This led to a compromise in the main objective of the attack. We tried averaging the similarity scores across multiple encoders. However, we found that while clean images maintained high similarity scores across all encoders, the adversarial images showed variability:
> ```
> Adv image   - 5 encoders similarity: [0.53, 0.73, 0.59, 0.82, 0.58]
> Clean image - 5 encoders similarity: [0.83, 0.89, 0.84, 0.81, 0.75]
> ```
>
> We have highlighted this part in the revised manuscript. We will upload it shortly.

---

### Official Review · Reviewer_fGhL · 2024-10-31

**Soundness:** 3
**Presentation:** 3
**Contribution:** 3
**Rating:** 6
**Confidence:** 2

**Summary:**

The paper proposes MirrorCheck, a novel method for detecting adversarial samples in Vision-Language Models (VLMs). It leverages a Text-to-Image (T2I) model to generate an image based on the caption produced by the victim model and then compares the embeddings of the input and generated images. The method is evaluated on different datasets and tasks, showing its effectiveness and superiority over baseline methods. It also demonstrates robustness against adaptive attacks and adaptability to different T2I models and image encoders.

**Strengths:**

- **Originality**: The idea of using a T2I model to generate images for adversarial sample detection in VLMs is novel. It provides a new perspective and approach compared to existing defenses.
- **Quality**: The experiments are well-designed and conducted. The use of multiple datasets, victim models, T2I models, and image encoders shows the robustness and generality of the method. The results are presented clearly and support the claims made in the paper.
- **Clarity**: The paper is well-written and organized. The introduction provides a clear motivation for the work, the method description is detailed and understandable, and the experimental results are presented in a logical manner.
- **Significance**: The proposed method has significant implications for the security and reliability of VLMs. It addresses an important problem of adversarial attacks and provides a practical solution that can be applied in real-world scenarios.

**Weaknesses:**

- The effectiveness of MirrorCheck depends on the quality of the pretrained generative model used. Any shortcomings in the generative model can directly impact the performance of the method. This limitation could be further explored and addressed.
- The paper could provide more in-depth analysis of the theoretical aspects of the method. For example, a more detailed explanation of why the image-image similarity approach works better than directly comparing the input image with the generated caption could be beneficial.

**Questions:**

- How can the method be further improved to handle cases where the pretrained generative model is of poor quality?
- Can the authors provide more insights into the theoretical foundation of the image-image similarity approach?
- Are there any potential limitations or challenges in applying MirrorCheck to other types of multimodal models or tasks?

---

> ### Author Response · Authors · 2024-11-22
>
> ```
> The effectiveness of MirrorCheck depends on the quality of the pretrained generative model used. Any shortcomings in the generative model can directly impact the performance of the method. This limitation could be further explored and addressed. How can the method be further improved to handle cases where the pretrained generative model is of poor quality?
> ```
> - We acknowledge that the performance of MirrorCheck is influenced by the quality of the pretrained generative model used. In our manuscript, we demonstrated its effectiveness using three state-of-the-art text-to-image models: Stable Diffusion, ControlNet, and UniDiffuser-v2. Our ablation studies confirm that the reliability and robustness of our defense pipeline are significantly enhanced when using well-trained, high-quality generative models. To further explore this limitation, we conducted additional experiments using less capable models i.e., Stable Diffusion with significantly reduced timesteps for image generation. **From the results in the table below**, while these models led to some performance degradation, MirrorCheck still maintained a reasonable level of effectiveness, underscoring the resilience of our approach even under suboptimal conditions. To improve the method's performance when using lower-quality generative models, future work could explore incorporating super-resolution techniques or fine-tuning smaller generative models to improve output quality. In practical applications, we strongly recommend service providers to prioritize high-quality generative models to maximize the robustness and reliability of MirrorCheck in real-world scenarios.
> | Victim Model   | Task | Setting       | RN50  | RN101 | ViT-B/16 | ViT-B/32 | ViT-L/14 | Ensemble |
> |:--------------:|:----:|:-------------:|:-----:|:-----:|:--------:|:--------:|:--------:|:--------:|
> | BLIP           | IC   | ADV-Transfer  | 0.870 | 0.850 | 0.825    | 0.890    | 0.780    | 0.843    |
> | Img2Prompt     | VQA  | ADV-Transfer  | 0.845 | 0.825 | 0.800    | 0.860    | 0.750    | 0.816    |
>
> ```
> The paper could provide more in-depth analysis of the theoretical aspects of the method. For example, a more detailed explanation of why the image-image similarity approach works better than directly comparing the input image with the generated caption could be beneficial.
> ```
> - In our work, we have detailed how our pipeline effectively counters adversarial attacks by leveraging an autoencoder-like mechanism to enhance robustness. This approach ensures that the defense operates collaboratively, improving the system's resilience to adversarial perturbations. We have expanded on these theoretical aspects in Appendix B, where we discuss the underlying principles and rationale behind our design choices.
>
> ```
> Can the authors provide more insights into the theoretical foundation of the image-image similarity approach?
> ```
> - The primary motivation for employing the image-image similarity approach stems from the inherent limitations of vision-language models (VLMs) in directly comparing input images with generated textual descriptions. **Research has shown** that VLMs often function as bags-of-words, struggling to effectively capture complex positional relationships and subtle variations in verb usage. These limitations can result in inaccuracies when performing cross-modal comparisons [1]. Our image-image similarity approach **addresses these issues** by leveraging the strengths of generative models, which are capable of producing semantically consistent images from textual descriptions. By comparing embeddings of the input and generated images, our method ensures a more reliable and precise assessment of similarity. This strategy minimizes the linguistic ambiguity and semantic loss that often occur in cross-modal transformations, providing a robust theoretical foundation for our defense mechanism. Furthermore, we selected this embedding-based similarity metric over conventional metrics like SSIM or FID because those methods **may fail to capture semantic equivalence** in cases where the Text-to-Image (T2I) model generates a visually different image that is still semantically similar. By utilizing vector embeddings, our approach maintains high similarity scores in such scenarios, ensuring robustness and reliability even when T2I outputs exhibit variability in their visual representation.
>
> - Reference: [1] Mert Yuksekgonul, Federico Bianchi, Pratyusha Kalluri, Dan Jurafsky, and James Zou. "When and why vision-language models behave like bags-of-words, and what to do about it." International Conference on Learning Representations, 2022.

---

> ### Author Response · Authors · 2024-11-22
>
> ```
> Are there any potential limitations or challenges in applying MirrorCheck to other types of multimodal models or tasks?
> ```
> - Many multimodal tasks, such as video captioning, rely on architectures that frequently incorporate frozen components like CLIP as an image encoder. This architectural similarity makes MirrorCheck adaptable for adversarial defense across various multimodal tasks, provided the models generate textual outputs. This flexibility ensures that our approach can generalize effectively to other domains beyond text-to-image scenarios. To **further validate the versatility of MirrorCheck**, we conducted experiments in unimodal settings (see Appendix C4), demonstrating its robustness and applicability across different use cases. Additionally, our method has been successfully tested against a completely different threat model—jailbreaking. Although MirrorCheck was not specifically optimized for this scenario, it outperformed other baselines, showcasing its adaptability and effectiveness under diverse adversarial challenges. **Please note that we are unable to add these results to comply with the review policy of ICLR**. These results highlight the potential of MirrorCheck to extend beyond its initial scope and address a broad range of multimodal tasks and threat models.

---

> > ### Author Response · Authors · 2024-11-27
> > **Grateful for your suggestions and awaiting further discussion**
> >
> > Dear Reviewer fGhL,
> >
> > Thank you for dedicating your time reviewing our paper and providing valuable feedback.
> >
> > We have thoughtfully addressed each of your comments, offering detailed responses to clarify and resolve the concerns you raised. We hope our explanations have provided a clearer perspective on our work and its contributions. These revisions have been incorporated into the manuscript, with the changes highlighted in blue for your convenience.
> >
> > If you feel that we have adequately addressed your concerns, we would be grateful if you would consider reassessing your rating.
> >
> > We would be happy to clarify or elaborate on any part of our paper while the discussion period is still open.
> >
> > Thank you,
> >
> > Authors

---

### Official Review · Reviewer_mrFP · 2024-11-05

**Soundness:** 3
**Presentation:** 3
**Contribution:** 3
**Rating:** 6
**Confidence:** 4

**Summary:**

The paper addresses the problem of adversarial defense for Vision-Language Models (VLMs). The paper proposes a simple and elegant method called MIRRORCHECK to detect adversarial samples. The method utilizes Text-to-Image (T2I) models to generate images based on captions produced by target VLMs. Then, the similarities of the embeddings of both input and generated images in the feature space are leveraged to identify adversarial samples. Extensive experiments are conducted to demonstrate the effectiveness of proposed method.

**Strengths:**

The proposed method is simple and elegant. It leverages Text-to-Image models to generate images based on captions produced by target VLMs. Then, the similarities of the embeddings of both input and generated images in the feature space are used to identify the adversarial samples.

The paper conducts sufficient experiments to demonstrate the advantages of proposed method. Specifically, the experiments are conducted on five victim models. More Text-to-Image models and pretrained image encoders are tested in the ablation studies.

The paper is well-written. For example, there is a section in Method to describe the threat model.

**Weaknesses:**

It is better to introduce how to choose hyperparameter \tau (decision threshold in line 217), especially the relation between \tau and text-to-images models (or pretrained image encoders).

The experiments are conducted against targeted attacks. How about untargeted attacks?

In Section 3.1, the paper writes "the attacker's objectives are to execute targeted attacks". If the attack performance of used attack method is low, will the proposed method still work?

There are some typos. For example, "Tables 1 and 3 presents" should be "Tables 1 and 3 present" in line 312.

**Questions:**

Though the paper conducts ablation study to explore the influence of different Text-to-Image (T2I) models and pretrained image encoders, is it possible to provide some information of T2I models such as classification accuracy and training mechanism to explain the relation between detection results and text-to-image models?

---

> ### Author Response · Authors · 2024-11-22
> **Rebuttal**
>
> We would like to thank the reviewer for the constructive review. We would like to answer all the concerns that have been raised.
>
> ```
> It is better to introduce how to choose hyperparameter $\tau$ (decision threshold in line 217), especially the relation between $\tau$ and text-to-images models (or pretrained image encoders).
> ```
> **Answer:** We determine the optimal value of $\tau$ using Receiver Operating Characteristic (ROC) curve analysis. Specifically, we identify the point on the ROC curve where the difference between the true positive rate (TPR) and the false positive rate (FPR) is maximized. This approach ensures a balanced trade-off between detection sensitivity and robustness, making  $\tau$ an effective decision threshold for identifying adversarial samples. Our experiments show that $\tau$ values vary but fall within a consistent range across different pretrained encoders. For the same encoder, $\tau$ values remain stable across different T2I models and attacked models. We will be adding this note to the final version of this paper.
>
> ```
> The experiments are conducted against targeted attacks. How about untargeted attacks?
> ```
> **Answer:** While the primary focus of our experiments was on targeted attacks, we also conducted additional experiments to evaluate the performance of MirrorCheck against untargeted attacks. These results are presented in Appendix C4, where we demonstrate that our method is effective across both attack types. Moreover, we have included further experiments to assess MirrorCheck’s robustness against Attack-Bard on **Image Description** (ID) task, an untargeted attack designed for VLMs [1]. The results are detailed in the table below. This comprehensive evaluation confirms the versatility and reliability of our approach under various adversarial scenarios.
>
> | Victim Model | Task | Setting      | RN50   | RN101  | ViT-B/16 | ViT-B/32 | ViT-L/14 | Ensemble |
> |--------------|------|--------------|--------|--------|----------|----------|----------|----------|
> | BLIP-2       | ID   | Attack-Bard  | 0.850  | 0.835  | 0.890    | 0.975    | 0.905    | 0.891    |
>
>
> ```
> There are some typos. For example, "Tables 1 and 3 presents" should be "Tables 1 and 3 present" in line 312.
> ```
> **Answer:** We will carefully review the manuscript to correct these, including the specific instance mentioned. We will ensure that all identified errors are addressed in the revised version.
>
> ```
> Is it possible to provide some information of T2I models such as classification accuracy and training mechanism to explain the relation between detection results and text-to-image models?
> ```
> **Answer:**  In our manuscript, we evaluated MirrorCheck using three state-of-the-art text-to-image (T2I) models: Stable Diffusion, ControlNet, and UniDiffuser-v2. Our extensive ablation studies in Figure 5 in the main manuscript demonstrate that our detection pipeline remains effective across different T2I models. Furthermore, we provide additional detailed results in Tables 12, 14, 16, and 18 in the Appendix. Across all experiments, we observe no difference in performance with changing the T2I encoder.
>
> ```
> In Section 3.1, the paper writes "the attacker's objectives are to execute targeted attacks". If the attack performance of used attack method is low, will the proposed method still work?
> ```
> **Answer:** Our method is designed to be effective regardless of the attack performance of the adversarial method used. If the attack method exhibits low performance, it may fail to generate adversarial examples that meaningfully alter the model's behavior. In such cases, the robustness of our defense may not be tested to its fullest extent, but our approach will still function as intended. Specifically, our defense mechanism is built to detect discrepancies between the input and generated representations, providing reliable protection even when the adversarial perturbations are less effective. MirrorCheck maintains its robustness across varying levels of attack strength. As seen in Figure 1 in the link provided below, with a very weak attack ($\epsilon=1$), the attack fails to generate adversarial examples that effectively alter the model’s behavior. As $\epsilon$ increases, the detection accuracy improves because the adversarial perturbations become more noticeable. However, with very high $\epsilon$ values (e.g., $\epsilon=32$), the images are almost destroyed, making them detectable even by humans.
> - Link: https://anonymous.4open.science/r/MC-6200/MirrorCheck_IMG.pdf
>
>  [1] Dong, Yinpeng, et al. "How Robust is Google's Bard to Adversarial Image Attacks?." arXiv preprint arXiv:2309.11751 (2023).

---

> > ### Author Response · Authors · 2024-11-27
> > **Grateful for your suggestions and awaiting further discussion**
> >
> > Dear Reviewer mrFP,
> >
> > Thank you for dedicating your time reviewing our paper and providing valuable feedback.
> >
> > We have thoughtfully addressed each of your comments, offering detailed responses to clarify and resolve the concerns you raised. We hope our explanations have provided a clearer perspective on our work and its contributions. These revisions have been incorporated into the manuscript, with the changes highlighted in blue for your convenience.
> >
> > If you feel that we have adequately addressed your concerns, we would be grateful if you would consider reassessing your rating.
> >
> > We would be happy to clarify or elaborate on any part of our paper while the discussion period is still open.
> >
> >
> > Thank you,
> > Authors

---

### Author Response · Authors · 2024-11-22

We would like to sincerely thank the reviewers for their thoughtful feedback. Your constructive insights have helped in enhancing the quality of our work, and we believe our paper is now significantly stronger. Below, we summarize our contributions and highlight the new additions to our work:

- **MirrorCheck Proposal:** We introduce *MirrorCheck* to detect and combat adversarial images targeting Vision-Language Models (VLMs). Our pipeline is model-agnostic and requires no additional training, leveraging pretrained models for this purpose.

- **Robustness to Untargeted Attacks:** While our initial focus was on targeted attacks against victim VLMs, we have shown that our pipeline is also robust to untargeted attacks. Furthermore, *MirrorCheck* can be applied to other domains, including classification and jailbreaking, showcasing its versatility.

- **Defense Against Adaptive Attacks:** We rigorously evaluate the strength of our defense against various strengths of adaptive attacks, ranging from grey-box to white-box scenarios, which represent the strongest forms of adversarial threats.

We will now proceed to address the specific questions and concerns raised by the reviewers. We will be adding all of these improvements to the final version of the paper. Once again, we deeply appreciate your valuable feedback and the opportunity to improve our work.

---

### Author Response · Authors · 2024-11-26
**Revised Manuscript**

Dear Reviewers,

We deeply appreciate the time and effort you have dedicated to reviewing our paper and providing your valuable feedback.

We have carefully addressed each of your comments and provided detailed responses to resolve the concerns raised. These revisions have been incorporated into the manuscript, with the changes highlighted in blue for your convenience.

We kindly invite you to review the updated manuscript, and if you feel we have satisfactorily addressed your concerns, we would be grateful if you could consider re-evaluating your rating.

If there are any additional questions or aspects requiring further clarification, please do not hesitate to let us know. We are more than happy to provide further explanations while the discussion period remains open.

Thank you once again for your thoughtful and constructive feedback!

---

### Author Response · Authors · 2024-12-03
**Thank you!**

Dear Reviewers,

We sincerely thank you for taking the time to review our submission. We deeply appreciate your efforts in evaluating our work and providing valuable feedback.

Our method's novelty and potential benefits to the community are substantial. We hope you will consider the contributions it offers in advancing VLMs security.

Your thoughtful insights and perspectives are invaluable to refining and improving our work, and we are grateful for your dedication to the peer review.

Best,

Authors

---

### Meta-Review · Area_Chair_nBR4 · 2024-12-19

**Metareview:**

This paper proposes a novel method to detect adversarial examples against vision-language models. The method adopts text-to-image models to generate images based on the captions produced by the target VLMs. Then the method calculates the feature similarities between the input image and generated image to identify adversarial examples. Experimental results show the effectiveness of the proposed method.

Overall, the idea is intuitive and novel. Utilizing a pre-trained text-to-image model to detect adversarial examples for vision-language models is interesting and is tailored for vision-language models rather than general defenses. The writing is good.

However, one major limitation of the method is its robustness under adaptive attacks. This issue has been raised by multiple reviewers and the authors have provided additional experiments to show the performance of adaptive attacks. After careful examination and discussion with reviewers, there are still some concerns.

- The authors has shown the results of attacking few CLIP encoders and using others to detect adversarial examples. This is **not** the white-box setting where the attackers can have access to all the defense details. So, using other CLIP models cannot prove the robustness of the method under strong adaptive attacks.

- Indeed, the authors also provided the results of attacking an ensemble of five CLIP encoders and using the same set of models for detection. However, the details on how to perform optimization on the adaptive attack problem are not clear. So, it is questionable whether the attack is strong enough to generate adversarial examples for the ensemble models.

As this paper has borderline rating with two borderline accept recommendations and two borderline reject recommendations, the AC thinks that this paper falls short of the ICLR acceptance threshold and can be further improved by carefully addressing more diverse attacks and adaptive attacks. Therefore, the AC would recommend rejection.

**Additional Comments On Reviewer Discussion:**

Below is the summary of the reviewer discussion.

- Reviewer mrFP initially raised concerns about chosen hyperparameters and untargeted attacks. The authors have adequately addressed the concerns.

- Reviewer fGhL initially raised concerns about limitation of using pretrained generative models and providing in-depth analysis of the method. The authors have adequately addressed the concerns.

- Reviewer RQhj initially raised concerns about unclear threat model, lack of defender capability, no theoretical analyses, insufficient robustness against adaptive attacks, and lack of experiments in OOD data and other baseline. After thorough discussions between authors and reviewer, most of the concerns have been addressed and the concern regarding adaptive attacks remains.

- Reviewer qvv1 initially raised concerns about robustness of the method, unclear experimental details, evaluation on adaptive attacks, etc.  After discussions, the reviewer still has concerns about adaptive attacks.

After carefully checking the reviews, author responses, and discussions, the AC believes that most of the concerns have been addressed but a major concern regarding adaptive attacks remains (see more details in the metareview). Therefore, the AC would recommend rejection.

---

### Decision · Program_Chairs · 2025-01-22

Reject